# Glycan-induced structural activation softens the human papillomavirus capsid for entry through reduction of intercapsomere flexibility

Yuzhen Feng[1,6], Dominik van Bodegraven [2,6], Alan Kádek [3,4,6], Ignacio L. B. Munguira[1], Laura Soria-Martinez[2], Sarah Nentwich[3], Sreedeepa Saha[2], Florian Chardon[2], Daniel Kavan [4], Charlotte Uetrecht [3,5] ✉, Mario Schelhaas[2] ✉ & Wouter H. Roos [1] ✉

High-risk human papillomaviruses (HPVs) cause various cancers. While type-specific prophylactic vaccines are available, additional anti-viral strategies are highly desirable. Initial HPV cell entry involves receptor-switching induced by structural capsid modifications. These modifications are initiated by interactions with cellular heparan sulphates (HS), however, their molecular nature and functional consequences remain elusive. Combining virological assays with hydrogen/deuterium exchange mass spectrometry, and atomic force microscopy, we investigate the effect of capsid-HS binding and structural activation. We show how HS-induced structural activation requires a minimal HS-chain length and simultaneous engagement of several binding sites by a single HS molecule. This engagement introduces a pincer-like force that stabilizes the capsid in a conformation with extended capsomer linkers. It results in capsid enlargement and softening, thereby likely facilitating L1 proteolytic cleavage and subsequent L2-externalization, as needed for cell entry. Our data supports the further devising of prophylactic strategies against HPV infections.

Human papillomaviruses (HPVs) are a large family of non-enveloped DNA viruses that infect squamous epithelia of skin or mucosa. HPV diseases range from asymptomatic infections to anogenital or oropharyngeal cancers, the latter of which are caused by so-called high-risk HPV types[1–3]. With an estimated 630,000 new annual HPV-related cancer cases, they have a huge impact on public health[4]. While type-specific prophylactic vaccines have been deployed to prevent HPV infection[5], additional strategies to fight HPV infections or related cancers remain in demand.

The life cycle of HPVs is closely linked to the differentiation of keratinocytes[6], where entry occurs in basal cells followed by amplification, replication and transformation in spinous cells, and attenuation of genome amplification and initiation of virus assembly in granular cells[7]. The icosahedral ($T = 7$ d) HPV capsids are assembled by the major and minor capsid proteins L1 and L2, respectively[8–10]. L1 forms homo-pentameric capsomers, of which 72 self-assemble into HPV particles[8,11]. Up to 72 copies of L2 are located mostly capsid-lumenal with a few surface-exposed peptides[12–14]. Capsid formation relies on

[1]Moleculaire Biofysica, Zernike Instituut, Rijksuniversiteit Groningen, Groningen, Netherlands. [2]Institute of Cellular Virology, ZMBE, University of Münster, Münster, Germany. [3]CSSB Centre for Structural Systems Biology, Deutsches Elektronen-Synchrotron DESY & Leibniz Institute of Virology (LIV), Notkestraße 85, Hamburg, Germany. [4]Institute of Microbiology of the Czech Academy of Sciences, Videnska 1083, Prague, Czech Republic. [5]Institute of Chemistry and Metabolomics, University of Lübeck, Ratzeburger Allee 160, Lübeck, Germany. [6]These authors contributed equally: Yuzhen Feng, Dominik van Bodegraven, Alan Kádek. ✉e-mail: charlotte.uetrecht@cssb-hamburg.de; schelhaas@uni-muenster.de; w.h.roos@rug.nl

the intercalation of L1 C-terminal arms of neighbouring capsomers. The C-terminal arms are proposed to dynamically sample different conformations, thereby contributing to capsid breathing[15]. This link between capsomers is stabilised by interchain disulfide bonds[16–20]. Due to the complex life cycle, culturing HPVs in vitro is similarly complex. As models for entry studies, virus-like particles (VLPs) and so-called pseudoviruses (PsVs) have been established. VLPs are formed by self-assembly of L1 or L1/L2 into empty particles, whereas PsVs are formed by L1/L2 VLPs encapsidating a chromatinised, reporter-gene expressing the pseudogenome[21,22].

HPVs bind to cells via the heparan sulphate (HS) moiety of heparan sulphate proteoglycans (HSPGs)[23,24]. Sequential engagement of distinct binding sites by HS located on the top rim of the L1-capsomer and in the cleft between two capsomers has been suggested[25,26]. Binding to HS but not to other highly sulphated glycosaminoglycans (GAGs) results in an ill-defined conformational change in L1 that we termed "structural activation", as exposure of L1 epitopes correlated with the gain-of-function to infect cells with undersulphated HS moieties when the virus is in close proximity[26,27]. This capsid alteration facilitates L1 cleavage by a secreted protease, most likely kallikrein-8 (KLK8)[27,28]. L1 cleavage is followed by cyclophilin-assisted externalisation of the L2 N-terminus from the capsid lumen[29,30]. Proteolytic cleavage of the L2 N-terminus by cellular furin reduces capsid affinity to HSPGs, likely followed by transfer to a secondary receptor[31–33]. Of note, the exact role of furin-mediated cleavage of L2, and whether it always requires cyclophin assistance has not been fully resolved[30,34]. Integrin α6, annexin A2 heterotetramer, growth factor receptors, and the tetraspanins CD63 and CD151 have been suggested as secondary receptor candidates[35]. The precise functions of the putative secondary receptors in orchestrating virus uptake remain, however, elusive. For example, triggering uptake may be facilitated by the formation of a dedicated microdomain of receptor candidates, a process perhaps assisted by matrix metalloproteinase-mediated cleavage of HSPGs[34,36,37]. The virus is internalised by a potentially novel endocytic mechanism[38,39], and routed via the endosomal pathway and retrograde transport to the Golgi apparatus[38,40,41]. Finally, the viral DNA is delivered to the nucleus after nuclear envelope breakdown during mitosis[42,43].

Capsid protein reorganisation, as proposed for HPVs, is common in the life cycles of various viruses[16,44,45]. However, the nature of these changes in HPVs with regard to structural alterations and their biochemical and mechanical consequences are so far unexplored. To address these complex issues at multiple levels, we combined virological approaches, atomic force microscopy (AFM), and hydrogen/deuterium exchange mass spectrometry (HDX-MS). While AFM nanoindentation enables the measurement of viral mechanical properties at a single-particle level[46], local changes of structure and dynamics in proteins can instead be mapped by HDX-MS[47,48]. Our results indicate that successful virion activation by HS binding requires a certain minimal glycan length. The glycan likely engages the virus at multiple binding sites and softens (i.e. making it easier to deform) the virus particle. Softening requires the engagement of HS binding sites in the cleft between virus capsomers and is reversible. Combined with our HDX-MS data showing decreased flexibility of both the N- and C-terminal arms of L1 connecting capsomers, we propose a molecular model of HPV structural activation. This model involves a pincer-like force exerted by heparin that locks the particle in the most extended of several alternating conformations of the invading C-terminal arm, which results in the enlarging and softening of the HPV capsid.

## Results

### Structural activation of HPV16 depends on glycan length

To address glycan requirements for what we termed structural activation, we made use of a facile seed over assay (Fig. 1A)[27]. In this assay, HPV16 PsVs are bound to the extracellular matrix (ECM)-resident laminin-332. Unperturbed cells that are seeded onto these ECM-bound viruses are infected, whereas cells with undersulphated HS moieties that are generated by NaClO3-treatment exhibit only background infection[27,49,50]. However, if the virus is activated by the engagement of HS or heparin, a fully sulphated HS analogue often used as an HS model, those undersulphated cells are well infected[27]. Short heparin oligosaccharides are able to bind to HPV16 capsids but fail to block infection during competition experiments, in which the virus is added together with GAG to cells, in contrast to long-chain heparin[27,51]. To date, it is unclear whether oligosaccharides would be able to activate the virus for infection of undersulphated cells as observed for long polysaccharides. Thus, we initially tested, if GAG length may be of importance for activation using heparin and HS of various lengths. NaClO3-treatment of HaCaT cells reduced HS sulphation of HaCaT cells by about 70%, abrogated infection by HSV-1, which depends on 3O-sulfation[52] and resulted in background infectivity of untreated HPV16 (Supplementary Fig. S1A, B and Fig. 1B, dotted line). Preincubation of HPV16 PsVs with long heparin chains bound to virus particles and restored infection of NaClO3-treated HaCaT cells in a dose-dependent manner, as expected (Fig. 1B and Supplementary Fig. S1C). Short heparin oligosaccharides that had a degree of polymerisation (dp) of five saccharides (Fondaparinux) or up to dp 20 (Fig. 1C, D) failed to do so. As short heparin oligosaccharides engage the virus[27,51], the results indicate that several HS binding sites on the capsid have to be simultaneously engaged by the same GAG to activate the virus for infection of undersulphated cells. As size-defined heparin oligomers longer than dp 20 are unavailable, we further probed entry using fractionated HS. Here, we used HS with a low degree of sulphation and an average dp 190 (dp190 lowS) as well as shorter HS with a high degree of sulphation and an average dp 40 (dp40 highS). HS dp190 lowS failed to activate the virus for infection (Fig. 1E). On the contrary, HS dp40 highS displayed a partial activation of infection (Fig. 1E) in line with partial activation. It is important to note that the reduced degree in activation of the HS dp40 highS fraction as compared to, e.g., heparin is likely the result of a lower degree of sulphation, and perhaps the inherent length dispersity of such fractions. Cumulatively, our results indicate that multivalent engagement of several capsid binding sites by one sulphated HS molecule of a length > dp 20 promotes activation.

### GAGs with longer saccharide chains trigger changes in the mechanical properties of HPV16

Next, we investigated whether there would be any mechanical consequences induced by HS engagement of HPV16. For this, we first probed changes in heparin-treated PsVs by AFM imaging and nanoindentation[53,54]. A single PsV was initially imaged by AFM, followed by pressing at its centre with the AFM tip (Fig. 2A). In this process, termed nanoindentation, the force detected by the AFM system and the tip indentation into the PsV was plotted as a force-indentation (F-D) curve. The viral spring constant, describing the particle's stiffness, was calculated by fitting the initial linear phase of the F-D curve, and the critical force was measured from the endpoint of this phase, indicating the maximum force prior to buckling or other non-linear deformations (Fig. 2B). AFM imaging revealed that heparin treatment significantly enlarged PsVs (Fig. 2C, D). Quantitative nanoindentation analysis showed that heparin engagement significantly decreased the viral spring constant and the critical force of PsVs (Fig. 2E and Supplementary Fig. S2A). These mechanical changes in PsVs, triggered by heparin, showed a heparin dose-dependent behaviour, which is consistent with the dose-dependent increase of infectivity described in Fig. 1.

Next, we asked whether the mechanical changes induced by HS engagement with HPV capsids would be dependent on glycan length. We used VLPs, which engage HS/heparin just like PsVs do[51], but as the VLPs lack DNA, any detectable changes in mechanical properties are solely due to changes in capsid conformation upon HS engagement.

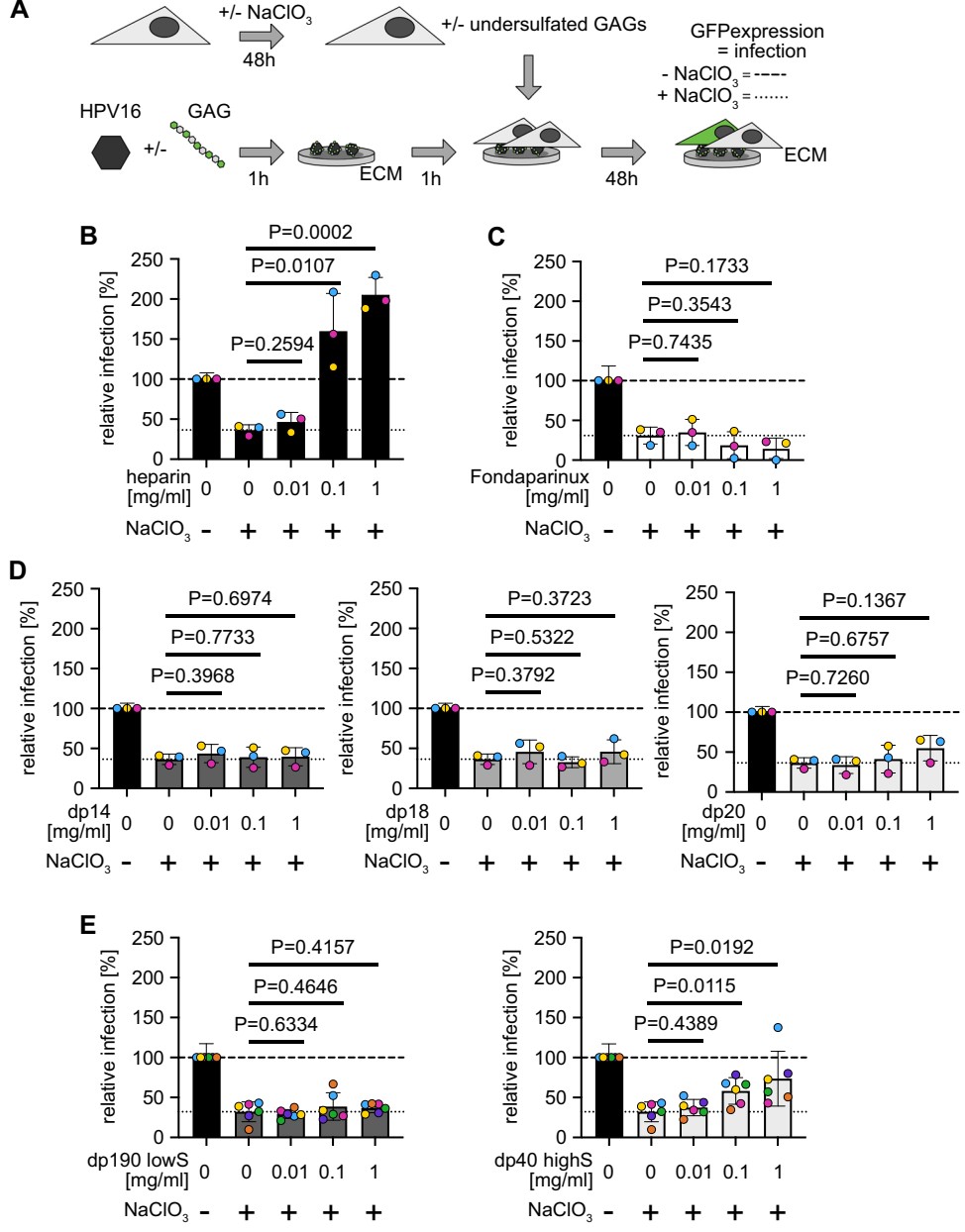

**Fig. 1 | Structural activation of HPV16 is dependent on GAG length. A** Seed over experimental schematic. Before binding to ECM, HPV16 PsVs were incubated with the indicated amounts of GAGs for 1 h. Untreated (striated line in **B**–**E**) or NaClO₃-treated (dotted line in **B**–**E**) HaCaT cells were seeded on top. Infection was scored by microscopy and displayed as the average of GFP-positive cells in % of total cells relative to untreated virus and cells as control for at least three independent experiments ± standard deviation (SD). The first column in each graph indicates untreated cells, where HPV16 was not incubated with GAGs to indicate the typical infectivity of cells. Error bar indicates here the error of normalisation to this column. Seed over was conducted with (**B**) heparin, (**C**) Fondaparinux, (**D**) short heparin oligomers or (**E**) heparan sulfate fractions. In (**E**), the average of six independent experiments is displayed ± standard deviation (SD). Striated lines indicate infection of untreated HaCaT cells with HPV16 PsVs not incubated with GAGs, and dotted lines indicate infection of NaClO₃-treated HaCaT cells with HPV16 PsVs incubated with GAGs at the given concentrations.

Heparin or Fondaparinux were used as poly- or pentasaccharides, respectively. VLPs retained their overall morphology irrespective of the presence or absence of glycans (Supplementary Fig. S3A). The quantitative assessment showed that Fondaparinux engagement did not significantly change HPV size and mechanical properties (Fig. 3C, D and Supplementary Fig. S4). Hence, mere binding of the longer glycan triggered changes in the capsid physical properties, which correlates with its ability to activate the particle.

Notably, the indentation-induced non-linear deformations were reversible in our nanoindentation measurements. There were no significant changes in the morphologies and sizes of HPV16 VLPs before

and after indentation (Supplementary Fig. S3A, B), indicating a high degree of particle robustness and flexibility withstanding loading forces up to 3 nN. The force curves corresponding to five successive indentations of a single particle showed that while the critical force slightly varied over five indentations, the slopes of each indentation were almost overlapping (Supplementary Fig. S3C). This suggests that the reversibility of HPV VLPs upon deformation results from its flexible and partially dynamic capsid structure where capsomers are interconnected by a network of linking peptides. In summary, we observed a dose-dependent softening of the HPV16 PsVs and VLPs after incubation with heparin but not Fondaparinux, as delineated by the

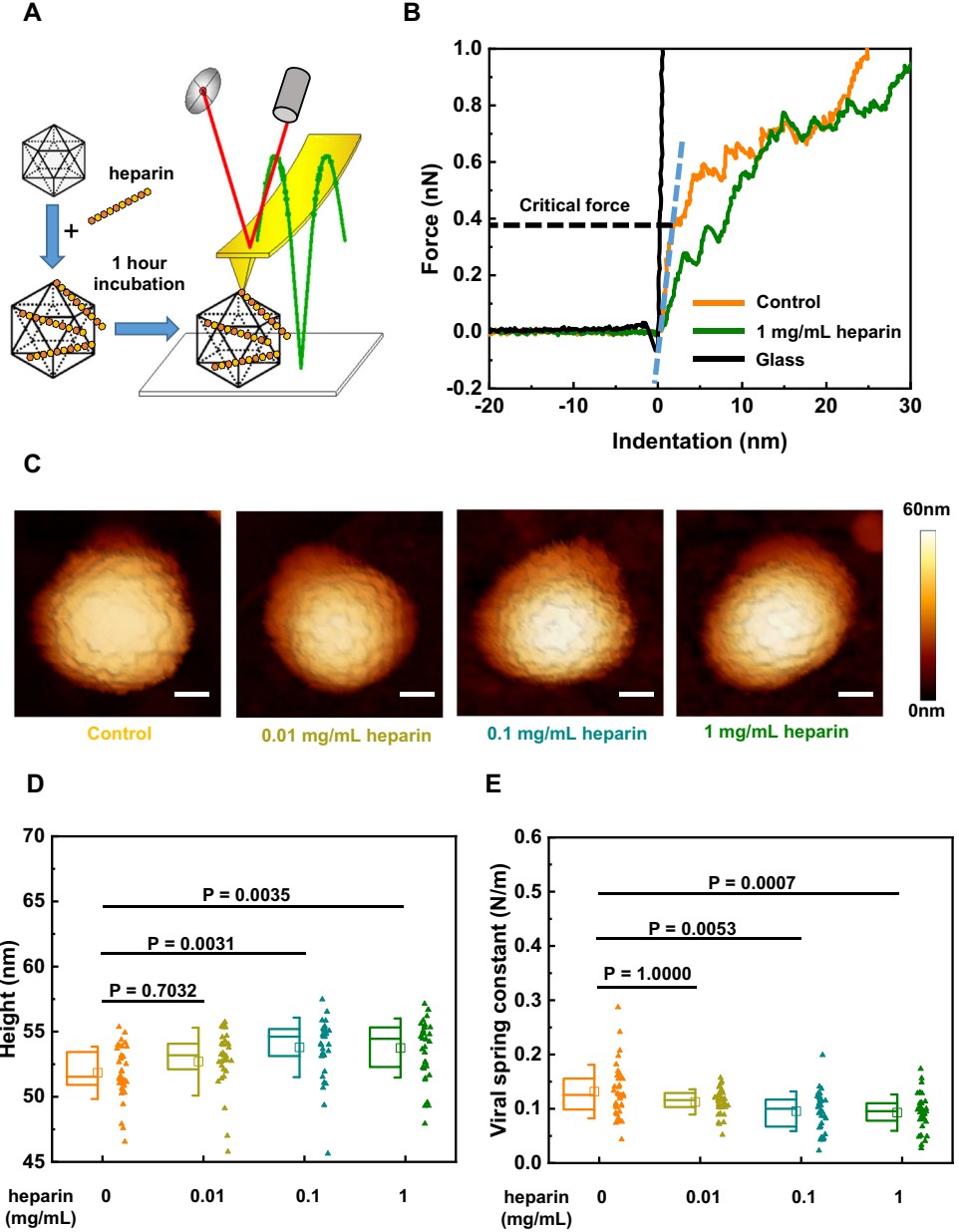

**Fig. 2 | Heparin triggers softening and enlargement of HPV16 PsVs in a dose-dependent manner. A** Schematic of the AFM setup. HPV16 PsVs were incubated with different concentrations of heparin for 1 h and then imaged and indented by AFM in liquid. **B** Typical force-indentation curves were acquired by using an AFM tip to indent a single HPV16 PsV under the indicated condition. The black curve was acquired by using an AFM tip to indent the glass substrate as a reference. The blue dashed line marks the linear phase on the approach line of the force-indentation curve used for linear fitting. The black dashed line represents the ordinate value at the endpoint of the linear phase, where the critical force of the particle was measured. **C** Typical AFM images of the PsVs incubated with the indicated concentrations of heparin before indentation. The scale bar is 20 nm. **D** Comparison of the height of the PsVs treated with different heparin concentrations. **E** Comparison of the stiffness of the PsVs treated by different concentrations of heparin. The significance level is 0.05 with Bonferroni correction for multiple comparisons. $P = 1.0000$ is due to Bonferroni correction. The number of data points (n) for the control group and the heparin-treated groups at concentrations of 0.01 mg/mL, 0.1 mg/mL, and 1 mg/mL are as follows: control group ($N = 36$), 0.01 mg/mL ($N = 35$), 0.1 mg/mL ($N = 34$), and 1 mg/mL ($N = 36$). The square in the box plots indicates the mean (also printed in Supplementary Table S1), the top and bottom of the box are the 25th and 75th percentiles, respectively, and the whiskers represent the standard deviation. The line in the box indicates the median. Source data are provided as a Source Data file.

observed decrease in viral spring constant. Since this correlates to infectivity observed in cell assays, activation is accompanied by structural changes, i.e. it is structurally activated.

**Heparin-binding induced HPV structural activation is reversible**
Since structural activation and the correlated softening of virus particles required GAGs with longer saccharide chains, we wondered whether multivalent interactions of one GAG with several sites on the viral capsid would be crucial to maintaining the conformational

change. To investigate the potential reversibility of structural activation, HPV16 PsVs were allowed to interact with immobilised heparin, subsequently eluted, and used to perform seed over experiments. Even though these PsVs were thus temporarily bound to heparin, no recovery of infection was observable (Fig. 3A). Likely, any structural activation triggered by transient binding to heparin failed to persist after elution.

To further corroborate this, heparin-bound HPV16 PsVs were subjected to heparinase treatment cleaving heparin into

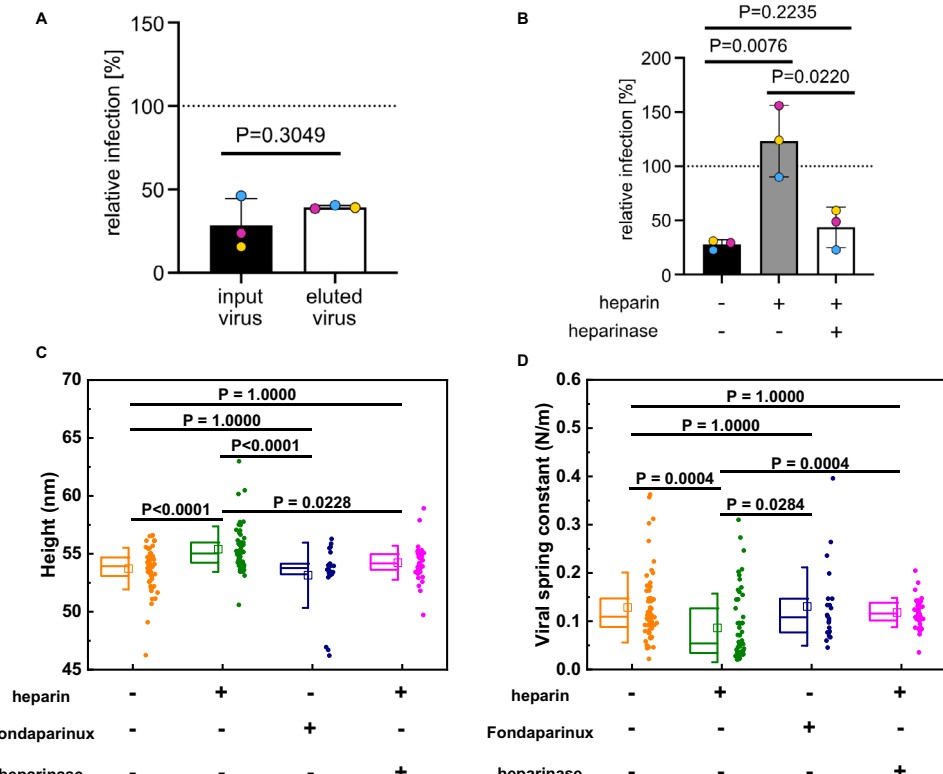

**Fig. 3 | HPV structural activation induced by heparin is reversible. A** Infection of NaClO₃ treated HaCaT cells with PsVs before (input) or after (eluted) being subjected to heparin affinity chromatography. $N = 3$ (**B**) Seed over experiment results of PsVs treated with 1 mg/mL heparin and the PsVs treated with 0.5 unit heparinase subsequent to heparin treatment. $N = 3$. Data are presented as bar centre (mean) ± error bars (standard deviation). Box plots of (**C**) the height and (**D**) the viral spring constant of VLPs show the reversible HPV structural activation induced by the longer GAG. To assess HPV structural activation reversibility, heparinase was applied subsequent to the heparin treatment. The significance level is 0.05 with

Bonferroni corrections for multiple comparisons. $P = 1.0000$ is due to Bonferroni correction. The exact $P$-values for the comparisons showing $P < 0.0001$ are as follows: $P = 3.8 \times 10^{-5}$ for the comparison between VLPs and heparin-treated VLPs, and $P = 4.5 \times 10^{-5}$ for the comparison between heparin-treated VLPs and Fondaparinux-treated VLPs. The square in the box plots indicates the mean (also printed in Supplementary Table S1), the top and bottom of the box are the 25th and 75th percentiles, respectively, and the whiskers represent the standard deviation. The line in the box indicates the median. Source data are provided as a Source Data file.

oligosaccharides. In contrast to heparin-bound HPV16, heparin-unbound and heparinase-treated viruses failed to exhibit structural activation, as evidenced by the loss of their heparin-associated recovery of infection of undersulphated HaCaT cells (Fig. 3B). Moreover, a similar reversion of mechanical properties and size of HPV16 VLPs after heparinase treatment was observed by AFM-based studies. Heparinase treatment led to similar viral spring constants, critical forces and sizes as untreated HPV16 VLPs, whereas heparin itself exhibited significant changes (Fig. 3C, D and Supplementary Fig. S4). This indicated that sustained interactions with long HS chains are indeed required to maintain the conformational change elicited by HS-virus interaction.

### K442 and K443 are essential for the activation of HPV16 through heparin

So, why would multivalent binding of GAG with longer saccharide chains be required to maintain a conformational change? Distinct HS binding sites were described in structural models from X-ray crystallography of HPV16 pentamer crystals soaked with a solution of heparin oligomers[25]. Intriguingly, these sites align and form a track that leads from the top rim into the canyon between two pentamers so that a multivalent binding of a longer polysaccharide along these sites is plausible (Fig. 4A). The binding site at the top rim (lysines 278 and 361) proposed to engage HS first[26] is furthest away from a site within the canyon between pentamers formed by lysines 442 and 443. The latter is directly connected to the invading C-terminal arm that intercalates

with the adjacent pentamer. Moreover, the epitope revealed upon structural activation (recognised by the H16.B20 antibody)[27] is located at the lower part of the invading arm. Thus, we hypothesised that a long heparin molecule needs to span several binding sites. As the binding sites lead from the top rim to the side of the pentamer, the GAG would be bent. Yet, highly sulphated HS or heparin are rather stiff molecules. The tension within heparin generated by binding to the capsid could thus generate enough force on the L1 molecule to induce a conformational change or stabilise an expanded conformation. Thus, the K442/443 site may work as a 'handle' for this tension-force generation as the heparin-binding site is directly connected to the invading arm, which is the likeliest site to result in biophysical changes if modified.

According to this model, removing the K442/443 binding site would result in decreased or absent structural activation. Hence, we generated K442/443A mutant HPV16 PsVs (HPV16 K442/443A). While these HPV16 K442/443A PsVs were similar to the WT in terms of the viral genome and minor capsid protein L2 incorporation (Supplementary Fig. S5B, D), particle morphology (Supplementary Fig. S5C), and viral mechanical properties (Supplementary Fig. S3D, E), they were unable to infect cells (Supplementary Fig. S5A). Loss of infection was mostly unrelated to cell binding, which was only slightly reduced compared to the WT, as expected (Supplementary Fig. S6A, B)[25,26]. To examine, if the mutant could still undergo structural changes during entry, we measured exposure of the L2 N-terminus upon infection by staining with an antibody against N-terminally located RG-1 epitope[29,55].

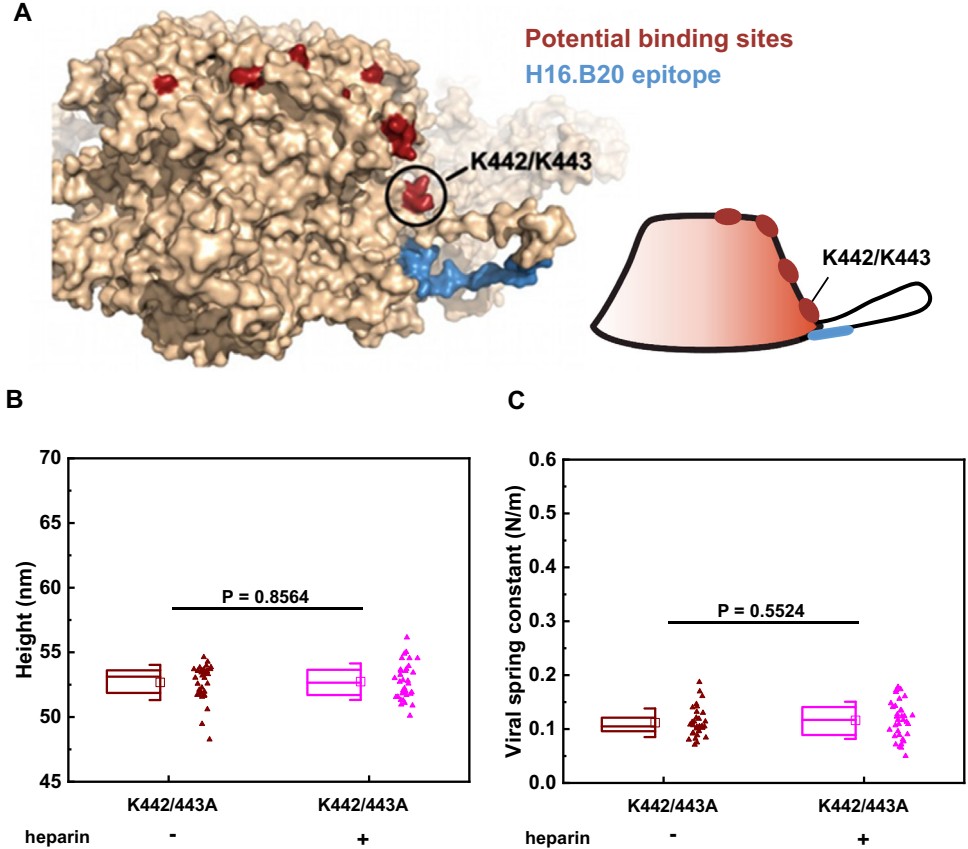

**Fig. 4 | K442 and K443 are essential for the activation of HPV16 through heparin. A** Visualisation of mutated site K442/443 on the L1 pentamer (pdb: 5kep)[13]. Putative binding sites are in red[25], and the epitope of H16.B20 antibody is in blue[83]. **B** Box plot of the height of the K442/443A PsVs incubated with ($N = 34$) or without heparin ($N = 34$). $P = 0.8564$. **C** Box plot of the viral spring constant of the K442/443A PsVs incubated with or without heparin. $P = 0.5524$. The significance level is

0.05. The engagement of heparin did not result in significant changes in size or stiffness in K442/443A PsVs. The square in the box plots indicates the mean (also printed in Supplementary Table S1), the top and bottom of the box are the 25th and 75th percentiles, respectively, and the whiskers represent the standard deviation. The line in the box indicates the median. Source data are provided as a Source Data file.

Indeed, the WT but not the K442/443A mutant PsVs exhibited RG1 epitope exposure upon infection indicative of altered structural capsid dynamics (Supplementary Fig. S6C, D). Using PsVs labelled with a pH-sensitive fluorophore that exhibits signals only upon exposure to the low pH of endosomes[56,57], we assessed the virus uptake. Both, WT and mutant internalised into endosomes as indicated by the fluorophore signal (Supplementary Fig. S6E, F). This indicated non-infectious uptake of the mutant, perhaps by a non-infectious pathway similar to previously reported interference with release from HSPGs or prevention of furin-cleavage of L2, which also leads to non-infectious internalisation[32,33]. Thus, the K442/443A mutant is unlikely to undergo structural activation upon HS engagement. Hence, we tested whether HPV16 K442/443A PsVs exhibited enlargement in size and changes in mechanical properties after heparin incubation. Importantly, this was not the case (Fig. 4B, C and Supplementary Fig. S2B), indicating that HPV16 K442/443A was not structurally activated, despite its ability to engage HS on cells (compare Supplementary Fig. S6A, B). Thus, most likely engagement of multiple HS binding sites on the capsid, including K442 and K443, is required for structural activation in line with our tension-force model.

## HDX-MS reveals heparin-induced stabilisation in the intercapsomer canyon

To better understand the structural consequences induced in HPV16 by heparin-binding, HDX-MS was used to probe the local conformational dynamics of PsVs. HDX-MS usually compares different states of a protein, e.g. with and without ligand, through monitoring the

spontaneous exchange of protein backbone hydrogens for deuterium in a $D_2O$-based buffer. This is observed by the mass increase of peptides, which are generated after the labelling reaction. The process probes the solvent accessibility as well as the involvement of amide hydrogens in hydrogen bonding and secondary structures. Changes in structural dynamics or engagement of amide hydrogens in ligand binding are hence reflected in increased (exposure / dynamics) or decreased (protection / stabilisation) HDX[47]. As heparin binding is largely driven by electrostatic interactions with side chains[58,59], the binding of short glycans will hardly be visible in HDX-MS. However, structural changes in the presence of heparin could alter the backbone hydrogen bonding. Therefore, PsVs were incubated in a $D_2O$-based buffer alone and in the presence of excess heparin (1 mg/ml), which induces full structural activation.

HDX-MS detected multiple L1 peptides of distinct protein regions with decreased deuteration upon heparin engagement (Fig. 5). The relatively small deuteration changes were, however, statistically significant and consistent across a number of peptides spanning the same protein region, thus conferring high confidence. Distinct populations or conformations in L1, e.g., in the pentavalent and hexavalent coordination, could, in principle, have different solvent accessibility resulting in bimodal deuteration distributions in HDX-MS spectra. However, as there are no signs of such behaviour on L1 in our dataset, we visualised the observed deuteration symmetrically over all quasi-equivalent copies of L1 in the structural model. Overall, most of the affected regions cluster towards both L1 N- and C-terminal regions (Fig. 5A). These are known to sample multiple conformations and form

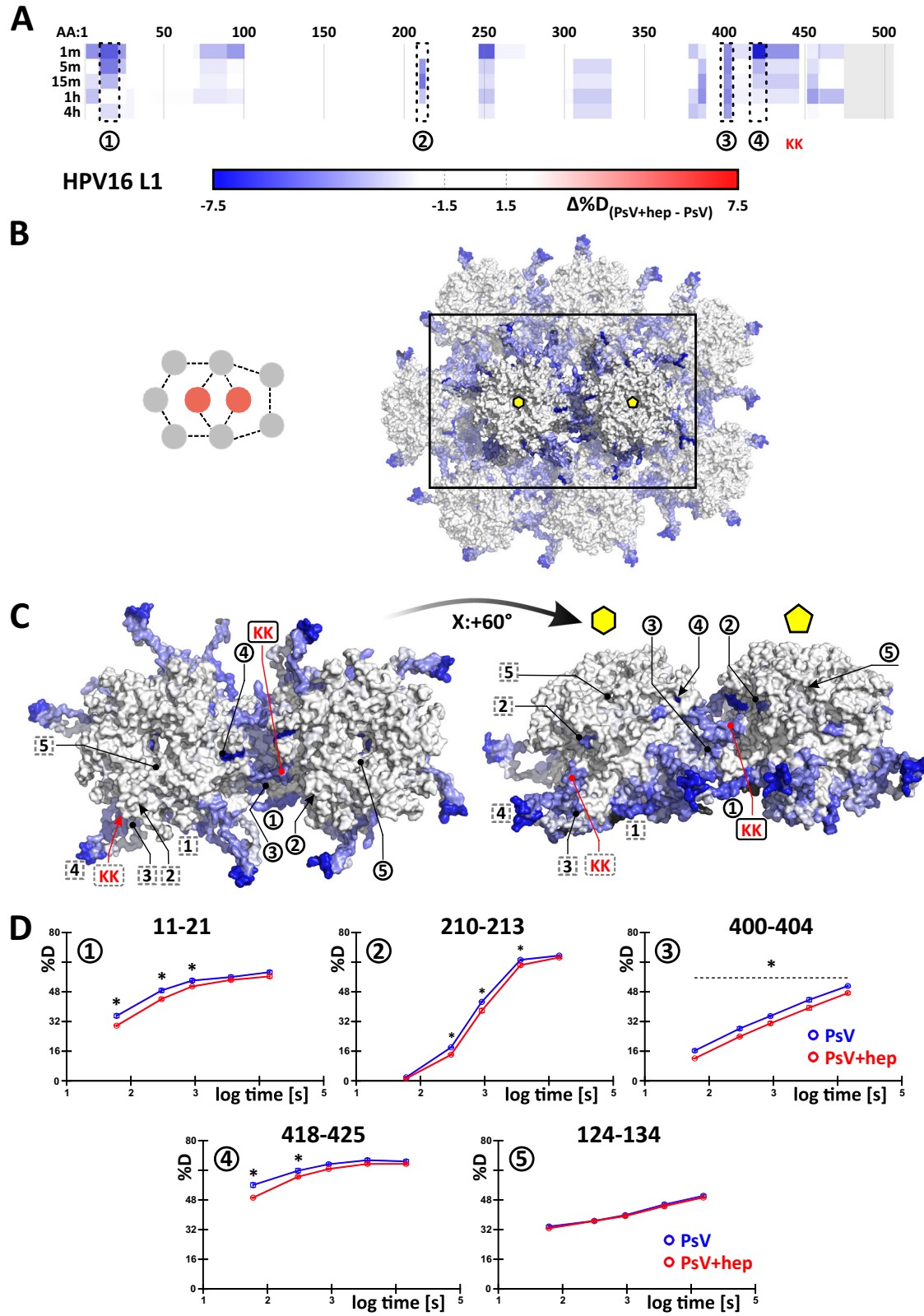

the base of the canyon linking adjacent capsomers[15]. Indeed, the maximal deuteration differences (Supplementary Fig. S7) superimpose predominantly with the intercapsomere canyon of HPV16 in a structural model (pdb:7kzf)[15] (Fig. 5B, C). The pentameric core of the capsomer remained mostly unaffected, which fits well its description as a largely rigid structure in a flexible environment of the pseudosymmetrical capsid[15]. Importantly, one of the regions most perturbed

by the binding of heparin was, in fact, the flexible C-terminal arm protruding into surrounding capsomers, the base of which contains the K442 and K443 residues involved in structural activation by heparin (Fig. 5C, red mark).

In principle, the decrease of L1 backbone deuteration might at least partially be attributed to lowered solvent accessibility caused by an obstructing heparin molecule positioned in the capsid canyon as

**Fig. 5 | HDX-MS shows lowered deuteration inside the intercapsomer groove in the presence of heparin.** **A** A subset of peptides describing all the observed changes during multipoint HDX-MS experiment are shown along the L1 protein sequence. Blue regions display decreased deuteration in the presence of heparin, and a grey area denotes missing data. Maximal observed differences are visualised symmetrically on the cryo-EM HPV16 structure (pdb: 7kzf). Capsomers in hexavalent and pentavalent coordination are shown in a broader context (**B**) and isolated (**C**). Regions for which example deuteration uptake plots are shown in (**D**) are marked on the asymmetric unit's chain "A" (forming the pentavalent capsomer) – black circles and on the asymmetric unit's chain "C" (part of the hexavalent capsomer), which is rotated about Z: +144° from the intercapsomer interface shown in (**C**) around the centre of the hexavalent capsomer - grey dashed squares. Black regions on the structure denote missing data in (**B** and **C**). Red marks the location of the K442/K443 patch, asterisk denotes statistical significance as described in the text. Region 5 shows an example peptide with no observed changes to exclude non-specific deuteration decreases across the whole particle. In (**D**) $N = 3$ replications (on the deuterium labelling level). The plots in (**D**) show mean value ± standard deviation with corresponding error bars mostly below the size of the point. The significant differences are determined by a two-tailed unpaired parametric $T$ test with a single pooled variance ($\alpha \leq 0.05$, with Holm-Sidak multicomparison correction). Source data are provided as a Source Data file.

modelled in previous cryo-EM data[13]. However, electrostatic heparin/HPV interactions are mediated by basic amino acid side chains instead of the peptide backbone[25]. Such side-chain-mediated interactions are often invisible in HDX-MS[58,59]. In fact, functionally verified HS binding sites on the rim and at the side of L1 capsomers[25] failed to exhibit a notable deuteration decrease (compare Fig. 4A and Supplementary Fig. 5C). Hence, the lowered deuteration more likely reflects decreased flexibility of L1 regions forming the canyon and corresponding rearrangement into more stable hydrogen bond networks upon structural activation.

In addition to L1, changes in the capsid-lumenal L2 would be plausible, since the L2 N-terminus must become exposed on the capsid surface for proteolytic processing and subsequent receptor switching[29,32,57]. Previous structural studies failed to conclusively identify L2 within the capsid, probably due to its variable occupancy and suspected flexibility[12,13,15]. In our HDX-MS experiments, certain changes were also observed on the lumenal side of L1 capsomers, i.e. a decrease of deuteration upon the exposure to heparin both in the L1 amino acid stretch 306–327, next to which a more stable and thus better-resolved density of a short stretch of putative L2 was previously identified in cryo-EM[15], and in the region 247–256, which forms the inner lining of the capsomer pore (regions A and B in Supplementary Fig. S8A, respectively). In line with changes on the lumenal capsid side, HDX-MS data exhibited a decrease of deuteration in several regions of L2, likely due to lowered flexibility of the protein upon heparin engagement, thus indicating that allosteric conformational changes indeed propagate to the L2 protein inside the capsid (Supplementary Fig. S8B). Notably, many regions without deuteration differences in L2 showed maximum deuteration within the first minute after labelling initiation (Supplementary Fig. S8C). This confirmed, through an experimental observation directly in a capsid assembly, previous computational and biochemical data on isolated L2 suggesting very high flexibility and inherent structural disorder in these parts of L2[15,60,61]. Further, in the N-terminal half of L2, bimodal deuteration distributions are found in HDX-MS independent of heparin treatment. These indicate the presence of two distinct conformations of L2 in the capsid or larger concerted movements of this part of the protein.

In summary, our HDX-MS data on the decreased exchange in L1 and L2 supports the relevance of the capsomer canyon and the invading C-terminal arm in heparin binding and structural activation. Importantly, this data is consistent with the model of flexible conformations of the L1 C-terminal arm[15], in which stabilisation of extended conformations by e.g. heparin engagement would result in a notable size increase of the capsid. This would stabilise a more structured form with stronger hydrogen bonding, hence resulting in the decrease in deuterium exchange observed.

## Discussion

In this work, we probed the structural activation of HPV16 virions for infectious entry upon HS engagement from different angles: Structural activation of virus particles for functional receptor switching and entry upon interaction with heparin correlated with particles becoming significantly softer. Both required the engagement of HS with longer polysaccharide chains and, thus, likely an engagement of several sites in the virus capsid by HS. As a structural consequence of HS engagement, primarily a stabilisation of the invading C-terminus of L1 into neighbouring pentamers was observed, likely in a more extended conformation, as this could account for the softening of the particle. Thus, we propose that the strain introduced by the binding of stiff HS provides the force needed to reach and stabilise an extended conformation of the C-terminal arm of L1 linking capsomers, which in turn would facilitate subsequent alterations of the capsid and eventually receptor-switching and uptake (Fig. 6). Counter-intuitively, the HS-induced stabilisation leads to a decrease in capsid stiffness. However, this stiffness decrease can be explained by the increase in size of the capsid, resulting in a more deformable particle.

Combining AFM nanoindentation, HDX-MS, and seed over assays, we demonstrated that structural activation by HS engagement is reversible and depends, both, on the length of the glycan and lysines 442/443 of the HPV16 L1 leading to stabilisation of the intercapsomer canyon formed by the L1 invading C-terminus with lysines 442/443 at its base. While most protein-HS interactions require glycan chains as short as dp 4, and while short heparin fragments can bind to the HPV16 capsid[27,51], short heparin fragments were insufficient for HPV16 structural activation. Together, this implies that the engagement of several binding sites by one HS molecule is crucial to induce or stabilise a particular conformation in the HPV16 capsid. No structural data on the engagement of long HS molecules to HPV16 capsids exists. Existing more indirect data, including this study, allows us to infer whether binding of one HS molecule occurs to several distinct HS binding sites within one L1 molecule or to binding sites spanning different L1 molecules: As it provides means to increase affinity/avidity for interactions, multivalent HS binding is not uncommon for viruses[62–64]. Theoretically, avidity might explain the lack of structural activation for HPV16 mutants, because low avidity binding of HS to capsids could result in low structural activation. However, since the binding of HPV16 K442/443A to cellular HSPGs was efficient (this study and[25,26]), this appears unlikely. Moreover, since structural activation required engagement of K442/443, it is more plausible that one HS molecule binds simultaneously to several if not all, binding sites in one L1 molecule in line with a previously proposed sequential engagement of binding sites[26]. Thus, we propose that a long stiff heparin/HS molecule binds to multiple binding sites on one L1 molecule, during which the heparin/HS molecule is bent and thus exerts forces on capsids. Here, the lowest binding loop, the lysines 442 and 443, would act as a handle for the GAG.

The residues 442 and 443 are directly adjacent to the invading arm of L1, which is crucial for particle stability and integrity by forming a covalently connected interpentameric network[16,65–67]. Indeed, HDX-MS data indicated a general decrease in deuterium exchange in this region. Since the invading arm is a rather flexible structure that can exist in several different conformations[15], HS engagement of K442/443 may stabilise one of those L1 C-terminal conformations in line with decreased deuteration. The different C-terminal arm conformations have been attributed to a phenomenon termed 'particle breathing', during which the L1 pentamers converge and diverge by about $7 - 9$ Å[15].

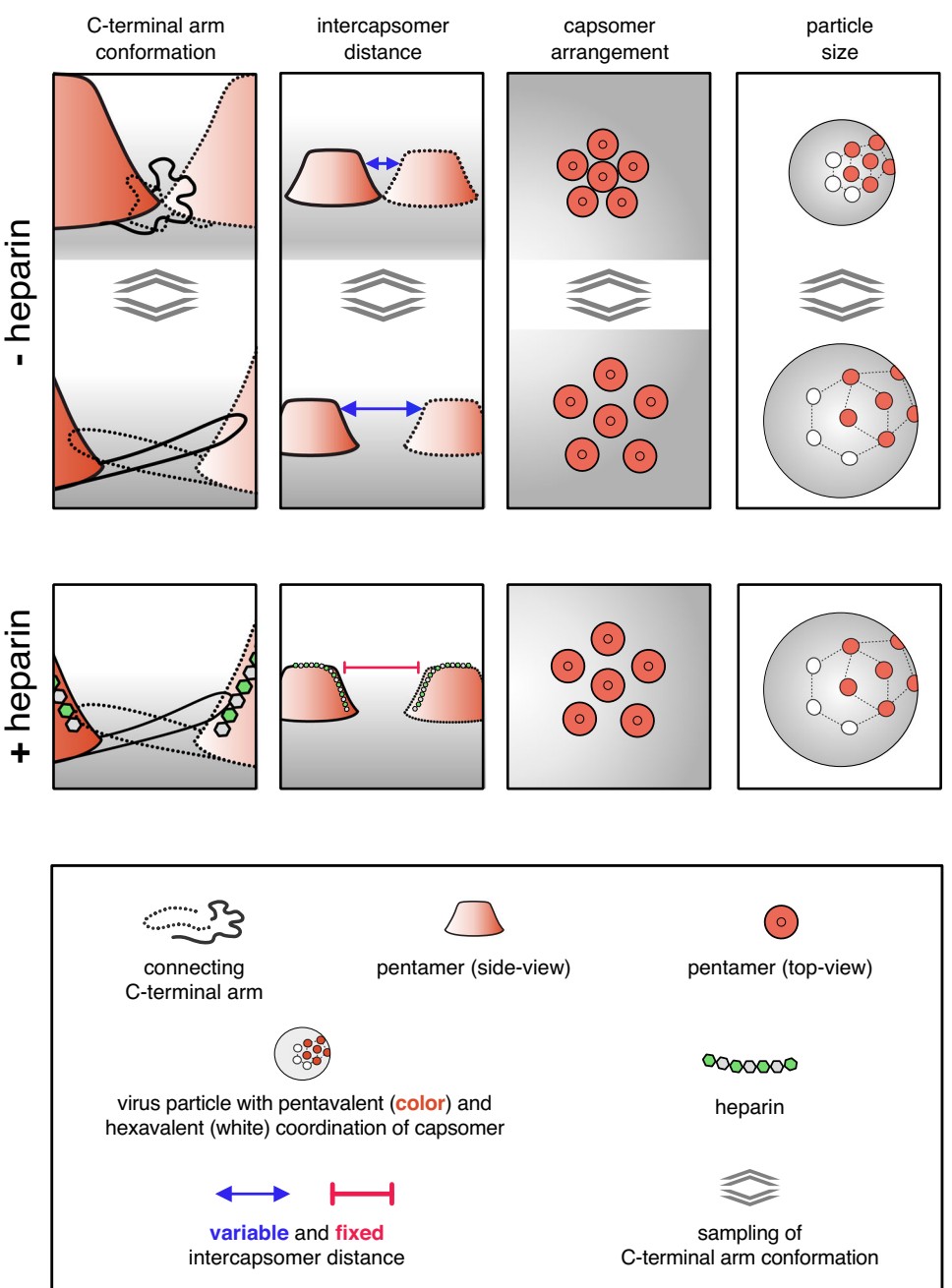

**Fig. 6 | Model of HPV16 structural activation through HS/heparin.** Schematic of HPV16 pentamer dynamics. The pentamers are viewed from the side in columns one and two and from the top in three. The last column displays particle size. Column one displays the invading arm that forms a disulfide bond in the adjacent pentamer to stabilise the particle. The invading arm is very flexible, which causes the pentamers to move towards and apart from each other (compare top and bottom), i.e. so-called 'particle breathing'. The binding of a long HS/heparin molecule to several aligned binding sites, including K442/443, stabilises the invading arm in the most extended conformation, which reveals the H16.B20 epitope as well as the KLK8 cleavage site. The maximum interpentameric distance leads to increased particle size and a softening in AFM, while additional hydrogen bonds in the stretched arms decrease HDX in this region.

If HS engagement would stabilise one of those L1 conformations, it would likely result in an observable change in the average diameter of the particles. Indeed, average particle size increased significantly upon engagement of long heparin polysaccharides but not of smaller oligomers (Figs. 2D, 3C)[13], suggesting that an extended conformation increasing the interpentameric distance was stabilised. An extended conformation of the L1 intercapsomer connection would also fit well with a softening of the virus particle, which was observed as a reduced spring constant in AFM (Fig. 2E). Moreover, the increased particle size is also in line with the enhanced accessibility of the HPV16.B20 epitope[27].

Overall, our mechanistic model for structural activation of HPV16 particles fits well within the paradigm of viral entry, in which crucial epitopes of viral structural proteins are exposed upon cell interactions in a timely and spatially exact manner to facilitate the next of many steps in the entry programme[68]. More specifically, our model may indeed explain how the immediate next steps in the entry programme would be facilitated: Subsequent to binding to HSPGs, L1 is cleaved by kallikrein proteases, a capsid-lumenal N-terminal L2 peptide is exposed to the virion surface by the help of cyclophilins, which is eventually cleaved by secreted cellular furin[28,29,32]. A stabilised L1 C-terminus within an enlarged particle would increase the efficacy of the crucial

kallikrein-8-facilitated proteolysis of L1 for cell entry[27,28]. In turn, both the enlarged capsomer distance and kallikrein cleavage of L1 would allow easier access of cyclophilins to the capsid lumen, promoting the exposure of the L2 N-terminus to the capsid surface, which would allow efficient furin-mediated cleavage of L2 that is crucial for receptor switching. It is also plausible that allosteric transmission of structural changes within L1 to the capsid lumenal L2 would help cyclophilin-mediated exposure of the L2 N-terminus. In an elegant way, HPV16 has evolved to stabilise one of several alternating conformations in the capsid to promote efficient entry. Making use of these findings by designing small compound inhibitors of such 'molecular gymnastics' and adding those, e.g., to lubricants may provide an additional strategy to reduce the burden of anogenital cancers.

## Methods

### Cell lines, plasmids, reagents and viruses
HeLa (CLL2) cells were from ATCC. HaCaT cells from N. Fusenig (DKFZ, Heidelberg, Germany)[69] and HEK293TT cells (CRL-3216)[21] were a kind gift from J. T. Schiller (NIH, Bethesda, USA). The plasmids pClneo-EGFP, p16L1h and p16Shell were kindly provided by C. Buck (NIH, Bethesda, USA)[22,70–72]. p16Shell K442/443A was generated by site-directed mutagenesis using the QuickChange II site-directed mutagenesis kit (Agilent) according to the manufacturer's instructions. For maintenance, cells were kept in Dulbecco's Modified Eagle's Medium (DMEM, Thermo Fisher Scientific), which was complemented with 10% foetal bovine serum (FBS, Biochrom). In the case of HEK293TT cells, the medium was supplemented with 400 μg/ml Hygromycin B. SYBR green master mix and BSA standard were purchased from Thermo Fisher Scientific. OptiMEM (Gibco), Lipofectamine 2000 (Invitrogen), Optical bottom 96w microplates (Greiner Bio-One). Recombinant HSV-1 expressing GFP (originally named HSV-1 17 CMV-IEproEGFP, here HSV-1-GFP) under control of the CMV immediate early promoter has been described[73] and was a kind gift of W. Hafezi (Münster, Germany).

### Virus preparation
HPV16 PsVs were prepared according to ref. 22. In short, p16Shell and pClneo-EGFP were transfected into HEK293TT cells. After 48 h, cells were harvested and lysed, followed by maturation of the PsVs for 24 h. For purification, the PsVs were subjected to a linear OptiPrep (iodixanol, Sigma-Aldrich) gradient (25–39% OptiPrep, 309600 × g, 5 h, SW60Ti rotor (Beckman Coulter). To generate the K442/443A L1 mutant virus, a p16Shell with the respective mutations was transfected together with pClneo-EGFP. Purification of VLPs was carried out accordingly, with the L1-only plasmid p16L1h but without the addition of the reporter plasmid pClneo-EGFP. For the purposes of HDX-MS analysis, the particles were purified using a CsCl step gradient (27% w/V and 38.8% w/V CsCl in 10 mM Tris-HCl pH 7.4, 207570 × g, 3 h 50 min, 4 °C) followed by dialysis in Float-A-Lyzer devices (1 mL, Spectra/Por) against a total of 3 L HPV virion buffer (1× PBS, 635 mM NaCl, 0.9 mM CaCl₂, 0.5 mM MgCl₂, 2.1 mM KCl, pH 7.4), since iodixanol, the main component of OptiPrep, precipitates in the presence of acetonitrile used for peptide liquid chromatography (LC) separation.

### Glycosaminoglycans
If not indicated otherwise, heparin (as sodium salt) was from Sigma-Aldrich (H4784, mainly dp45 – dp51). To investigate the influence of chain length on structural activation, a set of GAGs with a controlled length was utilised. Fondaparinux is a common clinically used heparin with dp5 (04191876, Viatris). The heparin oligomers dp14, dp18 and dp20 (HO14, HO18, HO20) and the HS fractions HS dp190 and dp40 (GAG-HS I and GAG-HS III) were from Iduron.

### HPV16 infection assays
About 4000 HeLa cells/well were seeded in 96-well optical bottom microplates 16 h prior to infection. To infect the cells, about $3 \times 10^7$

HPV16 PsVs were added and incubated at 37 °C for 2 h before the medium was exchanged and replaced by DMEM (10% FCS). Cells were fixed using 4% paraformaldehyde (PFA, in PBS) 48 h p.i. Nuclei were stained with RedDot2 (VWR). Cells were imaged with the 10x objective of a Zeiss Axio Observer Z1 spinning disc microscope (Visitron Systems GmbH), which was equipped with a prime BSI camera (Photometrics) and a Yokogawa CSU22 spinning disc module. For each of the three independent experiments, 32 fields of view were imaged per condition. Infection was scored as GFP-positive cells/ total cells in % in an automated manner using MATLAB-based InfectionCounter as previously reported[74]. Experiments for inhibition of infection by different glycans were performed by preincubating HPV16 and the respective glycan at RT for 1 h before infection of the cells was carried out as above.

### Analysis of NaClO₃ treatment of cells
Sodium chlorate treatment and analysis were carried out by treating HaCaT cells with 50 mM NaClO₃ for 48 h or performing the mock treatment, and subsequently, the cells were fixed in 4% PFA (in PBS)[27]. Sulfated HS moieties were stained using the A04BO8 antibody (kind gift of Toin van Kuppefelt, Nijmegen, The Netherlands) to detect mostly sulfated S-domains of HS[75] followed by incubation with anti-VSV antibody (Sigma-Aldrich), Alexa Fluor-488 labelled secondary antibody (Thermo Fisher Scientific), and phalloidin Alexa Fluor-647 (Thermo Fisher Scientific). Cells were imaged with the 63× objective of a Zeiss Axio Observer Z1 spinning disc microscope (Visitron Systems GmbH), which was equipped with a prime BSI camera (Photometrics) and a Yokogawa CSU22 spinning disc module. Signal intensity per cell was analysed by ImageJ (version: 2.14.0 /1.54 f).

Alternatively, untreated and sodium chlorate treated cells were infected with HSV-1 GFP at an multiplicity of infection of 2 plaque forming units/cell for 6 h. The number of infected (GFP-expressing) cells was determined by flow cytometry and normalised to the untreated control.

### Structural activation assay (seed over)
About 25000 HaCaT cells/well were seeded in 96-well optical bottom microplates 48 h prior to infection. Furthermore, 48 h before infection, different HaCaT cells were supplemented with 50 mM NaClO₃. This NaClO₃-treated medium was renewed every day until the cells were fixed. On the day of infection, the cells in the 96-well plates were incubated with 20 mM EDTA (diluted in PBS) for 45 min at 37 °C. Afterwards, they were gently removed, leaving the ECM behind. To test for their activation potential on HPV16, different glycans were incubated together with the virus for 1 h at RT. This mixture was added to the ECM and incubated for 1 h at 37 °C. About 4000 NaClO₃ treated or control HaCaT cells were seeded in the 96-well plates to be infected by the ECM-bound virus. 48 h p.i. The cells were fixed with 4% PFA (in PBS). Microscopy-based scoring of infection was executed as described above. To investigate if the structural activation is reversible, about 500000 cells/well were seeded in 12-well plates, but otherwise treated as described above. The virus/glycan mixture was additionally incubated with 0.5 U heparinase I and III (H3917, Sigma-Aldrich) for 1 h at RT before being added to the HaCaT ECM. About 50000 treated or controlled HaCaT cells/well were seeded to be infected by the virus. The cells were fixed 48 h p.i. and analysed for GFP expression by flow cytometry using a CytoFLEX S cytometer (Beckman Coulter). For the gating strategy, please refer to Supplementary Fig. S9.

### Analysis of heparin binding to HPV16 PsVs
Heparin-binding to HPV16 particles was analysed by preincubating 5 μg of HPV16 PsVs with increasing amounts of biotin-heparin (Sigma-Aldrich) or solvent in a total volume of 500 μl virion buffer[27]. Viruses were separated from free heparin by centrifugation at 310 000 × g for 5 h at 16 °C on an Optiprep step gradient (5%/39%). Viruses particles were collected from the interface between the two Optiprep

concentrations. The same amount of virus particles was bound to an ELISA plate overnight at 4 °C, and the amount of heparin was analysed using HRP-conjugated streptavidin (Pierce N-100). The amount of relative heparin binding was determined by absorbance readings of reacted HRP-substrate (TMB), subtraction of the blank, and normalisation to virus content.

## AFM sample preparation
For heparin treatments, the heparin (H4784, Sigma-Aldrich) was solved at a concentration of 50 mg/ml in 10 mM HEPES buffer (pH 7.4). HPV16 PsVs were incubated with the heparin solutions at final concentrations 0.01, 0.1 and 1 mg/mL for 1 h at room temperature before performing AFM measurements. An equivalent volume of 10 mM HEPES to the glycan solution was added to an HPV16 PsVs solution as the untreated control group. Glycan treatments on HPV 16 VLPs were performed accordingly, with Fondaparinux (04191876, Viatris) used as a short heparin oligosaccharide.

To enzymatically digest bound heparin, heparinase I and III Blend (H3917, Sigma-Aldrich) was solved with the concentration of 0.2 unit/ μl in the resuspension buffer, which was made of 20 mM Tris, 50 mM NaCl, 4 mM $CaCl_2$, 0.01 % BSA and adjusted pH to 7.5. Subsequent to incubating HPV16 VLPs with heparin, 0.5 unit of heparinase solution was added and incubated for another 1 h at room temperature. An equivalent volume of PBS to the heparinase solution was added to a heparin-HPV16 solution as a control group.

## AFM imaging and nanoindentation
Prior to AFM measurement, different HPV16 particle stock solutions were first diluted to a concentration of 11.3 μg/ml. Generally, AFM imaging and nanoindentation experiments were conducted following the previous protocol[53]. Briefly, a droplet of 100 μL HPV solution was deposited on a hydrophobic glass coverslip and incubated for 15 min at room temperature before adding another 1 ml PBS to a liquid receptacle. Both AFM imaging and nanoindentation were performed in liquid at room temperature using an AFM (Nanowizard 4 model, JPK). Rectangular cantilevers (qp-BioAC CB2, Nanosensors) with nominal spring constant at 0.1 N/m and nominal tip radii below 10 nm were used and calibrated using the thermal noise method. Before nanoindentation, a series of AFM images were collected using quantitative imaging mode at setpoint 50 pN − 80 pN to centralise one isolated HPV particle and ensure the target particle was in proper shape. After completing a high-resolution image of the target particle, nanoindentation was performed on the clean substrate neighbouring the particle to check AFM tip cleanliness. Then, the AFM tip immediately pushed the centre of the target particle at a loading velocity of 300 nm/s until 3 nN force was reached. Finally, the particle was imaged again to inspect the structural state after the indentation. AFM images were processed using JPK SPM data processing software (Version 6.1.163, JPK) and corresponding force curves were processed by Origin software. Statistical analyses were performed using SPSS software (ver. 24.0, IBM). Unless specified otherwise, *P*-values in the box plots were determined for height and critical force by one-way ANOVA test, and for viral spring constant by Kruskal-Wallis test as the corresponding dataset did not meet the assumption for one-way ANOVA test. Each individual data point in the box plots represents a separate experiment, i.e. on a different particle.

## Affinity-based purification of HPV16
About $1.5 \times 10^{12}$ of HPV16 PsVs were subjected to affinity chromatography on HiTrap Heparin HP (1 ml, VWR) using a NGC Quest System (BioRAD). While buffer A was composed of 10 mM sodium phosphate buffer pH 7.4, buffer B was additionally supplemented with 2 M NaCl. Sample application was performed in 15% buffer B, followed by a wash phase for 30 min (1 ml/min). Finally, the virus was eluted with a continuous 15–75% buffer B gradient and collected in fractions. Virus fractions were pooled and used in the seed over assay.

## Virus labelling
HPV16 PsVs were fluorescently labelled by incubating HPV16 with AF488, AF568, or pHrodo succinimidyl ester (Thermo Fisher) in a 1:8 molar L1:dye ratio in the dark for 1 h at RT while rotating[56]. Then, it was purified on an OptiPrep step gradient (15%, 25% and 39%) at 225884 × g for 2 h at 16 °C. The labelled virus appeared as a clear band and was collected. After the addition of 4% glycerol, it was snapfrozen in liquid nitrogen.

## Virus binding
One day prior to infection, 5000 HeLa cells were seeded on a coverslip. AF488-labelled HPV16 or HPV16_K442/443A PsVs was allowed to bind to the cells for 2 h. Afterwards, the cells were washed thrice with PBS and fixed in 4% PFA. Cells were permeabilized with 0.1% Triton X-100 for 15 min, followed by staining with phalloidin-AF647 (Sigma-Aldrich) in PHEM buffer for 30 min. Image stacks covering the whole cell were acquired with a spinning disc microscope (Zeiss Axio Observer Z1, equipped with a Yokogawa CSU22 spinning disc module; Visitron systems GmbH) using a 40× objective) Particles were counted with the 3D object counter of ImageJ (version: 2.0.0-rc68/15.2e)[76].

## Analysis of RG-1 exposure
To analyze the exposure of the RG-1 epitope, $1 \times 10^5$ HeLa cells were seeded overnight into eight-well microscopy chambers (Ibidi). Cells were infected with WT or K442/443A mutant HPV16 PsVs for 1 h when the inoculum was exchanged for growth medium. The cells were immunostained for the RG-1 epitope at 6 h p.i. Cells were set on ice, and the medium was replaced by ice-cold PBS. Cells were subsequently incubated with RG-1 and an AF488-labelled secondary antibody for 1 h, after which cells were fixed with 4% PFA (in PBS) and stained using phalloidin Alexa Fluor 647(Thermo Fisher). Image stacks were acquired using a spinning disc microscope (Nikon Ti2 eclipse equipped with a Dragonfly 600 spinning disc module, Oxford Instruments) using a 63× objective. Image analysis was carried out using using ImageJ (version: 2.14.0 /1.54 f). Values are reported as the RG-1 signal intensity per cell area.

## Virus internalisation
The internalisation of HPV16 PsVs was analysed as follows[56,57]. Firstly, HeLa ATCC were seeded at a density of 10^5 cells per glass coverslips. The next day, cells were infected with wild type or K442A/443 A HPV16-pHrodo. After 2 h, the inoculum was replaced with 1 ml medium and incubated for 4 more hours at 37 °C. At 6 h, image stacks covering the whole cell were acquired with a spinning disc microscope (Zeiss Axio Observer Z1, equipped with a Yokogawa CSU22 spinning disc module; Visitron Systems GmbH) using a 40× objective). Maximum intensity projections of the virus signal were created using ImageJ (version: 2.14.0 /1.54 f), and intensity-based analysis was quantified using CellProfiler V4.21. The total intensity of the internalised virus relative to cell number was normalised to the WT. Cell outlines were created manually for presentation purposes.

## VGE determination
Using 0.2 mg/ml proteinase K (Sigma-Aldrich) in HIRT buffer (400 mM NaCl, 10 mM Tris-HCl pH 7.4, 10 mM EDTA pH 8.0), DNA was released from HPV16 PsVs. To recover the DNA, a Nucleo Spin Gel and PCR Clean-up kit (Macherey-Nagel) were used, and the DNA was eluted in 20 μl TE buffer (10 mM Tris, 1 mM EDTA, pH 7.5). PClneo-EGFP incorporation was investigated by quantitative PCR (ABI7500, Applied Biosystems) using a plasmid standard to determine the number of

pseudogenomes. These were then related to the particle amount of the input.

## Determination of L2 incorporation

About 25 ng of WT and K442/443A HPV16 PsVs preparations were subjected to SDS-PAGE and Western blot analysis using L1 (Santa Cruz sc-476999) and L2 (Santa Cruz sc-65709) antibodies. Signals were acquired on an Intas ChemoTouch Imager (Intas, Germany), and signal intensities were quantified using ImageJ (version: 2.14.0 /1.54 f).

## HDX-MS analysis

HPV16 PsVs were pre-incubated for 1 h, either with or without heparin (H4784, Sigma-Aldrich) at room temperature. To initiate deuterium labelling, the samples were 6-fold diluted with the same buffer they were obtained in, only made of 99.9% $D_2O$ (150 mM NaCl, 4.8 mM KCl, 10 mM $Na_2HPO_4$, 1.8 mM $KH_2PO_4$, 0.9 mM $CaCl_2$, 0.5 mM $MgCl_2$, pD 7.2). This resulted in a final concentration of 0.5 μM L1 monomer in the form of PsVs with or without 1 mg/ml heparin during deuterium labelling. The exchange reaction was left to proceed at room temperature until aliquots of 45 μl were removed at predetermined time points (1 min, 5 min, 15 min, 1 h and 4 h). In the aliquots, the exchange was immediately stopped by twofold dilution with ice-cold quench buffer (0.25 M glycine, 100 mM TCEP, 8 M urea, indicated pH 2.7), resulting in a final pH of 2.5. Importantly, heparin, as well as DNA are highly negatively charged and thereby precipitate under HDX-MS quench conditions. Thus, we employed a strategy of Poliakov et al.[77], utilising protamine sulphate as an additive in the HDX quench buffer and modified it for HDX-MS with online proteolytic digestion. Therefore, for samples with heparin, the quench buffer additionally contained 1 mg/ml protamine sulphate (P4020, Sigma-Aldrich). After 30 s incubation on ice, the samples were centrifuged at $10.000 \times g$ for 1 min at 0 °C. Each supernatant was transferred to a fresh tube and flash-frozen in liquid nitrogen. Low-binding microtubes and low-binding pipette tips (both Axygen Maxymum Recovery) were used throughout for all handling of viral particles.

The frozen samples were quickly thawed and injected into a refrigerated (1 °C) HPLC system (Agilent Infinity 1260, Agilent Technologies), through a porcine pepsin column (≥ 3200 units/mg, Sigma-Aldrich) in-house immobilised[78] onto POROS-20AL perfusion resin (Thermo Scientific), which was kept at 4 °C. Pepsin digestion was performed at isocratic 200 μl/min flow rate (0.4% formic acid in water). After the digestion, peptides were online desalted for 3 min on a peptide microtrap (OPTI-TRAP, Optimise Technologies) and then eluted on a reversed-phase analytical column (ZORBAX 300SB-C18, $0.5 \times 35$ mm, 3.5 μm, 300 Å, Agilent Technologies). The LC separation proceeded at 25 μl/min flow rate through an 8 min gradient of 8–30% solvent B, followed by a 3 min gradient of 30–90% solvent B (solvent A: 0.4% formic acid in water, solvent B: 0.4% formic acid in acetonitrile). The outlet of the HPLC system was connected to an electrospray ionisation (ESI) source of an Orbitrap Fusion Tribrid Mass Spectrometer (Thermo Scientific). The instrument was operated in positive ESI MS-only mode for deuterated samples, scan range 300–2000 $m/z$, using 4 microscans at resolving power setting 120.000. In a separate measurement on a non-deuterated sample, the instrument was used in positive data-dependent ESI MS/MS mode with 30% HCD dissociation, 1 microscan and 240.000 resolving power setting for the identification of all peptides produced by non-specific pepsin cleavage.

In total 22 pmol and 50 pmol L1 protein were injected per MS and MS/MS analysis, respectively. To minimise sample carry-over on the protease column, two washing solutions were always injected between sample injections modified from Majumdar et al.[79] (wash solution 1: 5% acetonitrile, 5% isopropanol, 20% acetic acid; wash solution 2: 4 M Urea, 1 M glycine, pH 2.5). All HDX samples were analysed in technical triplicates, except for the 15-minute time point for PsVs without heparin, which was only measured in duplicate.

Peptides were identified from the MS/MS data by the Andromeda search algorithm implemented in MaxQuant (version 1.6.5.0) using a custom protein database containing the sequences of HPV16 L1 and L2 proteins. Deuterium uptake for the identified peptides was calculated with DeutEx (in-house developed), and manually inspected, and the statistical significance of the observed differences in deuteration was evaluated by applying an unpaired two-tailed Student's $T$ test with single pooled variance evaluated with alpha ≤ 0.05 using the Holm-Šidák correction for multiple comparisons in Prism 8.0.1 (GraphPad Software). Bimodal "EX1" HDX kinetics were identified by manual data inspection as well as by semi-automated peak broadening analysis at 25% peak intensity[80]. The processed data were visualised using MSTools (https://peterslab.org/MSTools/)[81] and open-source PyMol 2.6.0a0 (Schrödinger, Inc).

## Reporting summary

Further information on research design is available in the Nature Portfolio Reporting Summary linked to this article.

## Data availability

The HDX-MS data generated in this study have been deposited in the ZENODO repository with unrestricted open access (https://doi.org/10.5281/zenodo.10534050)[82]. The data from the virological assay and AFM measurement figures are available in the accompanying Source Data file. Source data are provided in this paper.

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

## Acknowledgements

C.U. and M.S. acknowledge funding through the Geman Research Foundation (DFG) within Research Group 'ViroCarb' (FOR2327): Glycans controlling non-enveloped virus infections' (UE 183/1–1 &–2, SCHE 1552/ 3-2). M.S. acknowledges additional funding through DFG within the Heisenberg-Programme (SCHE 1552 6-1) and for research equipment (INST 211/1029-1). W.H.R. and C.U. acknowledge funding through the EU ViruScan project (731868) and W.H.R. also through the EU INFRAIA consortium MOSBRI (101004806). A.K. gratefully acknowledges a postdoctoral research fellowship from the Alexander von Humboldt Foundation (1196583-HFST-P). The Leibniz Institute of Virology is supported by the Free and Hanseatic City Hamburg and the Federal Ministry of Health (Bundesministerium für Gesundheit, BMG). We are grateful to Hartmut Schlueter for access to the UKE proteomics core facility and the use of the Orbitrap MS.

## Author contributions

C.U., M.S. and W.H.R. conceived the project. Y.F., D.B., A.K., I.L.B.M., C.U., M.S. and W.H.R. contributed to the design of the experiments. Y.F. and I.L.B.M. conducted the AFM experiments and analysed AFM data. D.B., L.S.M., S.S. and F.C. produced HPV16 particles and performed virological assays. A.K., S.N. and D.K. took part in HDX-MS data collection and interpretation. Y.F., D.B. and A.K. drafted the first version of the article. C.U., M.S., and W.H.R. critically revised and rewrote the article. All the authors have approved the final version to be published.

## Competing interests
The authors declare no competing interests.
