## [Transparent Peer Review file · Nature Communications]

Glycan-induced structural activation softens the human papillomavirus capsid for entry through reduction of intercapsomere flexibility

Corresponding Author: Professor Wouter Roos

This manuscript has been previously reviewed at another journal. This document only contains reviewer comments, rebuttal and decision letters for versions considered at Nature Communications.

Version 0:

Reviewer comments:

Reviewer #1

(Remarks to the Author)

In this manuscript, a multidisciplinary team uses primarily biophysical approaches to study the interaction of glycans with human papillomavirus type 16 and its consequences on capsid properties and infection. Binding of HPV to heparin sulfate proteoglycans has long been recognized as a key initial binding step during infection, but the molecular details are obscure. They report here that heparin binding not only mediates cell binding but also induces capsid enlargement and “softening” as well as decreases flexibility of parts of the L1 major capsid protein. They propose that these events facilitate proteolytic cleavage of L1 and exposure of the L2 minor capsid protein, which is required for proper trafficking of the viral DNA to the nucleus. Shorter oligomers are inactive in these assays. These are interesting observations on an understudied aspect of HPV infection (which causes a significant number of human cancers), and they provide important mechanistic new insight into the initial phases of infection. However, there are several concerns with the current manuscript that need to be fixed (some of which require only a simple experiment).

Major comments:

1. The most serious problem is that they perform their AFM measurement on capsid deformation on virus-like particles, VLPs, consisting solely of L1. By contrast, authentic virus (as well as pseudoviruses, PsV, which they use to measure infectivity) contain not only L1 but also L2, as well as encapsidated DNA associated with nucleosomes. I think it is likely that these differences might be important. For example, the presence of chromatin tightly packed in the particle might well make it more rigid. So, their AFM and infectivity assays in response to various glycans are comparing apples to oranges. This problem is not resolved in Figure S2, which claims, “HPV PsV and 37 K442/443A PsV have similar mechanical properties as HPV VLP”, but they don't look at the effect of heparin, which is the whole point of the manuscript!
2. A second problem is that they don't know what step of infection is blocked by short glycans or by the mutations at the lysines. Reporter gene readout is a very downstream measure of a long and complex process, whereas they claim that their manipulations interfere with the conformational changes required for successful infection. They should use the antibodies they mention to confirm the absence of the conformation changes, and confocal microscopy (or some other measure) to confirm absence of virus internalization.
3. There is also a potential problem with the lysine mutants. Do those mutants package L2? If not, they will be non-infectious.
4. Do the shorter glycans induce the changes seen with HDX-MS?
5. The data showing capsid enlargement is clear cut and easy to appreciate. I suggest moving it from the supplemental data to the main figures. Is “height” the right term to use, rather than diameter?

Minor comments:

1. The Hafenstein lab has reported previously flexibility in HPV capsids. Although one of the papers is referenced (#49) in terms of the L1 capsomer structure, it should be cited in the introduction as a prior report of HPV capsid flexibility.
2. The figures of the structures would be easier to comprehend if they were complemented with simple cartoons.
3. A brief description of the AFM measurements, as was done for the HDX-MS, would be helpful for the non-biophysicist. What do capsid softening, critical force, and spring constant mean?

4. Regarding the difficulty in visualizing L2 and its apparent flexibility, there is now ample biochemical, genetic, and computational published evidence that much of L2 is unstructured (PMID: 30375341, 37819982).
5. Related to major comment 1 above, the authors are imprecise in discussing the actual capsids they are studying. For example, in the methods line 558, they say “HPV16” when they mean HPV16 PsV; and on line 576 they say “HPV16 particles” when they mean HPV16 L1 VLPs. I can understand using shorthand at times, especially in the main text, but certainly in the methods they should be precise!
6. In Figure S5, I assume the changes are in response to heparin addition, but this is not stated in the figure legend.
7. I would remove the big X and upper row of panels in the + heparin panels in Figure 6.
8. There are some problems with the English. For example, the last sentence of the abstract is garbled; on line 111, insert “generated” after e.g.; line 220, replace “not conserved” with “did not persist”; line 358/359 “Regions whose example deuterium uptake plots...”??

Reviewer #2

(Remarks to the Author)

The manuscript by Feng et al. present a structural study of the papillomavirus capsid as it interacts with heparan sulfates. Much of the functional assays, hydrogen deuterium exchange with MS, and AFM studies support the idea that binding to larger and highly sulfated heparan chains is necessary to induce structural rearrangements. The manuscript is well-written, logical, and presents mostly reasonable conclusions. There are a few minor issues that need to be addressed prior to acceptance.

- 1) In addition to sharing the raw spectra data, the authors should include a summary spreadsheet file of the fitted H/D exchange data as suggested by Masson et al (Nature methods): <https://doi.org/10.1038/s41592-019-0459-y>. It is hard to evaluate the overall consistency of the data without having simple access to the exchange data peptides beyond the few that are shown in the main figures.
- 2) There is a slight concern that adding heparin, which is highly acidic, to the sample will slightly decrease the solution pH and may cause perturbations similar to what is observed among the peptide uptake plots presented (slight protection everywhere). Did the authors take this into account? Adding another example peptide uptake plot to the figures that shows gradual uptake over time and shows no change with heparin would alleviate this concern.
- 3) The authors conclude that protection is altered uniformly across all of the capsid subunits. However, could the L1 and L2 changes occur only in a subpopulation that is immediately proximal to the heparin binding sites. You may have very strong protection, but only at a portion of the capsid subunits, and when doing the centroid analysis for deuterium uptake, this averages out to a relatively small observed change in protection. Or is binding to any surface of the capsid causing a universal change in protection across all of the capsid subunits? Is there any evidence of peak width broadening at the peptides that are the most perturbed that would indicate that the protection is not uniform across the capsid? (See Weis et al., doi: 10.1016/j.jasms.2006.05.014). The authors should at least mention something related to this in the results or discussion.

Reviewer #3

(Remarks to the Author)

The authors combined virological assays with hydrogen/deuterium exchange mass spectrometry, and atomic force microscopy to investigate the effect of capsid-HS binding and structural activation. They found that there is a HS-induced structural activation that requires a minimal HS-chain length and simultaneous engagement of several binding sites by a single HS molecule. They also explored the nature of structural changes of HPV during maturation. The results suggest a specific length of glycan is required to engage the virus at multiple binding sites, which leads to decreased flexibility. Although the exploration of virus events during entry is pertinent, they are the only group to describe (in a 2013 paper) what they are calling “structural activation” and there is a need for quantification.

Other concerns:

Entry events are not as straightforward as what is described in the third paragraph of the introduction since there are many conflicting reports about the process.

It is unclear what is meant by soften or softening, Line 97 introduction and again at the end of introduction: “softening of the HPV capsid.” “...softens the virus particle. Softening requires engagement...”

Using “soft” to describe a virus capsid is ambiguous and cannot be quantified.

That heparin can exert a “pincer-like force” seems highly speculative and unlikely.

The authors state that the virus is structurally activated by HS or heparin to become infectious and that they will test the length required to “structurally activate.” This term from their 2013 publication is used throughout the manuscript without explanation. Example: Short heparin failed to activate.

A claim is also made that the results indicate that several HS binding sites on the capsid have to be simultaneously engaged “to structurally activate the virus for infection.” There should be provided the ratio of how many HS were incubated per

capsid. More troubling is the lack of any explanation of what sort of structural change is required before the virus is infectious. How is the “activation” possible if the capsids were infectious prior to HS incubation?

Line 271 states that tension within heparin generated by binding to the capsid could generate enough force on the L1 molecule to induce a conformational change and that K442/443 site may work as a handle for this tension-force generation. This model is highly speculative. A tension-force generation seems unlikely.

The HDX-MS results reported in Figure 5 are not significant differences. This section is over-interpreted.

Most of the results section seems over-interpreted.

Reviewer #4

(Remarks to the Author)

The authors present an interesting method to address the effect of glycosaminoglycans on the mechanical properties and infectivity of the human papillomaviruses. The work employs a variety of techniques and methods, which are well described, to identify the conformational changes induced in the HPV16 virion by the heparan sulphates. The findings suggest that heparin binding induces structural changes in the L1 arm inducing an extended conformation. This increases the capsid size and reduces the spring constant of the particle, as measured by AFM. The enlargement of the capsid facilitates cleavage of the L1 capsid protein, L2 externalization, triggering cell entry. They show that a minimal chain length of HS is required to activate the conformational changes in the virion, as demonstrated by an increase in infectivity and softening of the capsid when the capsid is exposed to heparin and other heparan sulphates of increasing length. The authors present an interesting and novel approach to characterize the nanomechanical and structural changes during the initial stages of virus, which are significant to the field. However, before publication, clarifications and additional information are required to support the article's conclusions.

1) In fig 1a the scheme presented where the HaCaT cells are treated with NaClO₃ is not clear. The authors could consider adding additional information to the scheme to clarify the treatment applied to the cells before seeding. The authors should demonstrate that NaClO₃ treatment is effective by performing an experiment to demonstrate sulfation expression and if possible, to quantify the decrease in sulfation?

2) In fig 1B-1E the individual datapoints of the results should be displayed. The authors also use dotted lines to show the infectivity of untreated cells (shown at 100%). As the graph shows relative infectivity to untreated cells, setting this line to 100% is not very informative and probably misleading as it does not show the variability of infectivity of untreated cells. It would be more interesting to also have the variability of the individual values for these untreated cells displayed as a first bar graph with the individual values. Was this variability taken into account in the statistical tests? For the NaClO₃-treated cells, the data should also be presented in the form of individual points, mean and standard deviation, as is done for the different heparin concentrations.

3) In fig 1c&1d the label reads “heparin [mg/ml]” instead of the concentration of Fondaparinux, dp14, etc.

4) Figure 2c shows the surface VPLs after incubation with heparin. Heparin is a molecule with a considerable size of several nm. Could the authors demonstrate the presence of heparin on the surface of the virions from the topographical images? Are all virions covered with heparin? Is the amount of heparin the same for all virions? This is important because only a small number of virions have been analyzed by AFM and these should be representative of the population. The only evidence for the presence of heparin is the increase in virion radius, but the difference is minimal and not statistically significant. To validate the presence of heparin and fondaparinux and to quantify their amount at the level of individual virions, the authors should provide an alternative method, such as fluorescence/immunofluorescence analysis, to label and empirically measure the density of molecules bound to individual virions.

5) In Figure 2D and E, the amount of datapoints varies between 60 and 22 data points. This may not be enough to support the conclusions of the article. Are these separate experiments? The authors should also provide a measurement of the spring constant of the same virion before and after adding heparin an/or fondaparinux, to avoid the effect of changes in the cantilever between experiments or other differences in the measurement conditions. To extract the constant stiffness of the virus, the authors subtract from their experiments the contribution of the stiffness of the AFM lever. How can the heparin or Fondaparinux present between the AFM tip and the virions influence the extracted stiffness constant? In other words, how can the authors be sure that the decrease in stiffness constant is not due to brush-forming heparin molecules on the surface of the virus, rather than a change in the organization of the virus particle?

6) In fig 2B a typical FC is displayed for each experiment. The curves show that under the same 3 conditions, the response is linear until approximately 7 nm of deformation. Is this a common characteristic of the virion and how it is affected by presence of heparin?

7) The authors show in the infectivity experiments that dp40-highS and lower concentrations of heparin also influence the HPV16 activation. To prove the effect of the chain length and the progressive changes induces in the virus, AFM measurements of the spring constant of HPV capsids under lower concentration of heparin and dp40-highS should be provided. The same concentrations as those used in figure 1 should be tested by AFM to validate the hypothesis that the change in infectivity following an increase in heparin concentration comes from a progressive change in the mechanical stability of viral capsids.

8) The authors state that “non-linear deformations were reversible in our nanoindentation experiments.” While I agree with these results in the linear regime, non-linear forces and the value of critical force is different between consecutive approaches, as shown in figure S2.

9) In figure 3C and 3D the data presented is the same as in figure 2D and 2E. The authors should consider rearranging the way information is presented to avoid unnecessary repetition of results within different figures.

10) During the measurement of the spring constant of the mutated HPV16, the spring constant of the mutants is presented

with only 21 points. More independent experiments might be required to support the results presented.

11) In fig 6 no labels appear in the picture. The authors should also improve the scheme's quality and add additional information to facilitate understanding of the model presented. In addition, the third panel is not necessary for the presentation of the model and should be removed from the main text.

12) In the supplementary material, the cross section along the diagonal of the virus is provided. The measurement of the cross section along the direction of the scan should be provided instead, as it is less prone to be affected by imaging artifacts. It would also be interesting to provide the height of the virus as a separate measurement showing the change in height under the presence of the HS. As mentioned above, more data and statistical analysis should be presented here to assess the relevance of these data.

Version 1:

Reviewer comments:

Reviewer #1

(Remarks to the Author)

The authors have responded adequately my all of my concerns.

Reviewer #2

(Remarks to the Author)

The authors have addressed my main scientific concerns. My only remaining issue is with the way the summarized data have been made available. Zenodo requires a login to access the pertinent data, which for the purpose of manuscript review is not appropriate. It would essentially reveal my identity to the authors. I applaud all efforts to make more data available, but in this case (and in the future) the authors should simply upload a summary of the HDX-MS results with the manuscript or deposit the data where it is readily available without credentials. I am only specifically referring to the spreadsheet with the summarized results and sampling conditions as outlined in the Nature Methods HDX guidelines paper. File size should not be a factor, as the summarized results of even the largest HDX-MS data sets would not exceed tens of megabytes.

Reviewer #3

(Remarks to the Author)

There are no further comments.

Reviewer #4

(Remarks to the Author)

I would like to congratulate the authors on the excellent revisions made to the manuscript. The additional experiments and thoughtful revisions have significantly enhanced the quality and clarity of the paper.

I do, however, have a few minor suggestions for further improvement.

1/ In Figure 1, I noticed that the reference column for relative infection is clearly presented in the reply to the reviewers, but it should also be included in the article itself. Additionally, I recommend that the authors use the same normalization method for all datapoints in this figure and display the error used for normalization.

2/ In Figure 2D, the new experiments measuring the spring constant at varying heparin concentrations provide valuable insights and nicely illustrate the gradual change in the spring constant. However, I had also requested that this measurement be conducted using dp40-highS, but the results for this are missing.

Aside from these points, I find the paper to be of high quality, with interesting results that offer promising perspectives for future research.

Reviewer #5

(Remarks to the Author)

Reply to Reviewer comments

Reviewer #1 (Remarks to the Author):

In this manuscript, a multidisciplinary team uses primarily biophysical approaches to study the interaction of glycans with human papillomavirus type 16 and its consequences on capsid properties and infection. Binding of HPV to heparin sulfate proteoglycans has long been recognized as a key initial binding step during infection, but the molecular details are obscure. They report here that heparin binding not only mediates cell binding but also induces capsid enlargement and “softening” as well as decreases flexibility of parts of the L1 major capsid protein. They propose that these events facilitate proteolytic cleavage of L1 and exposure of the L2 minor capsid protein, which is required for proper trafficking of the viral DNA to the nucleus. Shorter oligomers are inactive in these assays. These are interesting observations on an understudied aspect of HPV infection (which causes a significant number of human cancers), and they provide important mechanistic new insight into the initial phases of infection. However, there are several concerns with the current manuscript that need to be fixed (some of which require only a simple experiment).

Major comments:

1. The most serious problem is that they perform their AFM measurement on capsid deformation on virus-like particles, VLPs, consisting solely of L1. By contrast, authentic virus (as well as pseudoviruses, PsV, which they use to measure infectivity) contain not only L1 but also L2, as well as encapsidated DNA associated with nucleosomes. I think it is likely that these differences might be important. For example, the presence of chromatin tightly packed in the particle might well make it more rigid. So, their AFM and infectivity assays in response to various glycans are comparing apples to oranges. This problem is not resolved in Figure S2, which claims, “HPV PsV and 37 K442/443A PsV have similar mechanical properties as HPV VLP”, but they don’t look at the effect of heparin, which is the whole point of the manuscript!

Reply: We acknowledge the reviewer's concerns on how the VLP AFM results link to the infectivity assays and what the effect of heparin is on PsV as measured by AFM. In order to address this concern, we performed additional AFM measurements on HPV16 PsV treated with different concentrations of heparin. Consistent with our findings in HPV16 VLPs, heparin engagement induced a dose-dependent softening and enlargement in HPV16 PsV (Figure 2D-E). These results align with the in vitro infectivity assays, thereby validating the relevance of our AFM measurements to the infectivity assays. Regarding the effect of L2 and pseudo-genome incorporation on the mechanical properties of HPV capsids, we did not find a significant difference (Figure S3D-E).

2. A second problem is that they don't know what step of infection is blocked by short glycans or by the mutations at the lysines. Reporter gene readout is a very downstream measure of a long and complex process, whereas they claim that their manipulations interfere with the conformational changes required for successful infection. They should use the antibodies they mention to confirm the absence of the conformation changes, and confocal microscopy (or some other measure) to confirm absence of virus internalization.

Reply: We apologize for the misunderstanding: Short heparin oligomers bind the HPV16 particle but do not block infection of cells (Dasgupta et al., 2011; Knappe et al., 2007, Cerqueira et al., 2013) or activate the particle (this study). It has been proposed that long chain molecules display a higher affinity compared to shorter oligomers due to avidity effects, i.e. engagement of several binding sites on the virus particle. This could lead to exchange of short heparin oligomers and engagement of long chain HS molecules on cells. We have now mentioned this in the results section at lines 146-149.

For the lysin mutant, we added two experiments to additionally address this issue. For one, we assessed whether the mutant is able to expose the RG1 epitope during infection, a conformational change just downstream of HS binding (Cerqueira et al., 2015). Of note, RG-1 epitope exposure was abolished for the mutant indicating a change in the structural dynamics of the capsid (Figure S6C-D). Second, we measured the degree of internalization of the mutant in comparison to WT virus. Since the mutant particles were internalized with a slight yet mathematically insignificant increase and a tendency for more perinuclear accumulation, the mutant was taken up non-infectiously. We would speculate that uptake occurs by a different mode leading to degradation rather than infection (Figure S6E-F). Since similar findings have been reported when impairing furin-cleavage of L2 or preventing receptor-switching (Selinka et al., 2007; Richards et al., 2006), this data is consistent with impairment of the structural alterations that lead to receptor switching and infectious uptake.

3. There is also a potential problem with the lysine mutants. Do those mutants package L2? If not, they will be non-infectious.

Reply: Failure to incorporate sufficient amounts of L2 into mutant PsV would indeed reduce infectivity, but would likely also lead to reduced amounts of pseudogenome incorporation (Wang, Roden 2013), which we did not observe (Figure S5B). Here, we provide additional evidence that the mutants package L2 in similar amounts to WT HPV16 PsV (Figure S5D-E) strongly indicating that the loss of infectivity is not the result of assembly defects.

4. Do the shorter glycans induce the changes seen with HDX-MS?

Reply: Based on the seedover assays, in which short glycans fail to recover infectivity, we expect only binding and no long-range changes in HDX-MS. This is further supported by cryo-EM data from Hafenstein's group (Guan et al., 2017). They have shown electron density in the canyon for bound short heparin glycans. Moreover, heparin binding mostly involves electrostatic interactions with side chains instead of the peptide backbone, hence it would be hard to trace sole binding of the short heparin glycans in HDX-MS. This assumption is further corroborated by the invisibility of functionally verified binding sites in the PsV-heparin dataset. As pseudovirus material for HDX-MS was limiting, we focussed on long heparin molecules, which clearly induced a structural change. Such larger changes rearrange the backbone and should be reflected in HDX-MS. We discuss the results in light of the published cryo-EM data in the main body of the manuscript as well as the reasoning to focus on long glycans (line 693-696).

5. The data showing capsid enlargement is clear cut and easy to appreciate. I suggest moving it from the supplemental data to the main figures. Is "height" the right term to use, rather than diameter?

Reply: We have moved the height graphs of the HPV particles to the main text, as suggested by the reviewer (Figure 2D, Figure 3C and Figure 4B). In AFM typically "height" instead of "diameter" is used to describe the size of particles because of the convolution effect between the AFM tip and the measured particle during imaging. This causes the "diameter" (i.e. size in the x-y direction in an AFM image) in the AFM cross-profile to be wider than the actual size of the particle. Therefore, "height" provides a more accurate measurement of the particle size in AFM experiments.

Minor comments:

1. The Hafenstein lab has reported previously flexibility in HPV capsids. Although one of the papers is referenced (#49) in terms of the L1 capsomer structure, it should be cited in the introduction as a prior report of HPV capsid flexibility.

Reply: We have cited the suggested reference in the introduction section at line 70-71 where we added the following sentence: "The C-terminal arms are proposed to dynamically sample different conformations, thereby contributing to capsid breathing."

2. The figures of the structures would be easier to comprehend if they were complemented with simple cartoons.

Reply: Thank you for your suggestion. We have added cartoons in Figure 4A and Figure 5B to enhance comprehension.

3. A brief description of the AFM measurements, as was done for the HDX-MS, would be helpful for the non-biophysicist. What do capsid softening, critical force, and spring constant mean?

Reply: We agree this is useful and, a brief description of the AFM measurements has been added to the results section at line 213-221. Furthermore, the first time soft is mentioned in the introduction we now added the notion of deformation. This now reads: "and softens (i.e. making it easier to deform) the virus particle." at line 128.

4. Regarding the difficulty in visualizing L2 and its apparent flexibility, there is now ample biochemical, genetic, and computational published evidence that much of L2 is unstructured (PMID: 30375341, 37819982).

Reply: We agree – these results go in line with our experimental observations of L2 flexibility inside the pseudovirus capsid presented here. While not the main focus of our story, we have introduced the above references into the manuscript.

5. Related to major comment 1 above, the authors are imprecise in discussing the actual capsids they are studying. For example, in the methods line 558, they say "HPV16" when they mean HPV16 PsV; and on line 576 they say "HPV16 particles" when they mean HPV16 L1 VLPs. I can understand using shorthand at times, especially in the main text, but certainly in the methods they should be precise!

Reply: We agree with the reviewer's comment regarding the need for precisely describing the actual capsids being studied. We have revised the text, figure legends, and methods sections using either PsV or VLPs to ensure clarity in describing the specific types of HPV16 particles in each experiment.

6. In Figure S5, I assume the changes are in response to heparin addition, but this is not stated in the figure legend.

Reply: Indeed, we changed the figure caption (Fig S8 in the revised manuscript) as well as added a separate HDX difference scale bar to clarify this explicitly.

7. I would remove the big X and upper row of panels in the + heparin panels in Figure 6.

Reply: Thank you for your suggestion. We have adapted the model of Figure 6 to make it clearer and followed your advice to remove the third row of panels.

8. There are some problems with the English. For example, the last sentence of the

abstract is garbled; on line 111, insert “generated” after e.g.; line 220, replace “not conserved” with “did not persist”; line 358/359 “Regions whose example deuterium uptake plots...”??

Reply: We appreciate the reviewer's detailed comments. The corresponding sentences have been revised accordingly.

Reviewer #2 (Remarks to the Author):

The manuscript by Feng et al. present a structural study of the papillomavirus capsid as it interacts with heparan sulfates. Much of the functional assays, hydrogen deuterium exchange with MS, and AFM studies support the idea that binding to larger and highly sulfated heparan chains is necessary to induce structural rearrangements. The manuscript is well-written, logical, and presents mostly reasonable conclusions. There are a few minor issues that need to be addressed prior to acceptance.

- 1) In addition to sharing the raw spectra data, the authors should include a summary spreadsheet file of the fitted H/D exchange data as suggested by Masson et al (Nature methods): <https://doi.org/10.1038/s41592-019-0459-y>. It is hard to evaluate the overall consistency of the data without having simple access to the exchange data peptides beyond the few that are shown in the main figures.

Reply: Due to their size, we have provided the community-recommended HXMS overview table as well as all the exported data points both in graphical uptake plots as well as in the tabulated form in the raw data open deposition at ZENODO as referred to in the data availability section of the manuscript. The HDX-MS overview table describing the experiments was, in light of this comment, also moved directly into the supplementary information to be more readily available. For your convenience, we also provide the reviewer link here: [- 2\) There is a slight concern that adding heparin, which is highly acidic, to the sample will slightly decrease the solution pH and may cause perturbations similar to what is observed among the peptide uptake plots presented \(slight protection everywhere\). Did the authors take this into account? Adding another example peptide uptake plot to the figures that shows gradual uptake over time and shows no change with heparin would alleviate this concern.](https://zenodo.org/records/10534050?token=eyJhbGciOiJIUzUxMiJ9.eyJpZCI6IjU3YjMyMWZjLTY0NWEtNDgyOS1hM2RkLWYzYTU4YjEyMTNINSIsImRhdGEiOi9LCjYyYm5kb20iOiJiYzRhZDdhNWl1ZGMxOTZlZlkODUyN2ExMTQ1MDFiOSJ9.Mo3UTt68iTP6DrFPiXOoI0HDDw9kaAkDmG1C-s5tHiEM2qV-e_qyZ-qAcaJooRuqBKDhu2F3CqZyp4qhsrgIOA; <https://doi.org/10.7554/eLife.37295>). (ii) To address the concern regarding the varying number of data points, we conducted a random sampling of 40% of cases (using SPSS software, ver. 24.0, IBM) from the control and heparin groups, ensuring the number of data points in these groups closely matched those in the Fondaparinux-treated group. Statistical analysis of these samples showed a similar trend for heparin and Fondaparinux effects on HPV capsids, with heparin increasing capsid size and decreasing capsid stiffness, while Fondaparinux exhibited no significant effects on capsid size and stiffness. Please see the figure below for this data.

For both these reasons, we think our approach is valid. Regarding the measurement of the same viral particle before and after adding heparin or Fondaparinux, this approach is challenging, especially when statistics need to be built up. In addition, viral particles must first be immobilized on a substrate for AFM imaging and indentation. This immobilization prevents full engagement of heparin or Fondaparinux with the entire viral particle, as the portion in contact with the substrate cannot interact with these molecules.

Above we already discussed the increase in size of the particles after

addition of heparin (see new fig. 2). This increase is small (but significant) and measures 1-2 nm as follows from figure 2D (see also Table S1). As can be seen in panel B of the same figure, the linear phase of indentation including the end of the first force drop extends roughly 5 nm, so much more than the increase in size induced by heparin. So this indicates that we are measuring the stiffness of the HPV capsid itself rather than the stiffness of a heparin brush layer above the capsid.

- 6) In fig 2B a typical FC is displayed for each experiment. The curves show that under the same 3 conditions, the response is linear until approximately 7 nm of deformation. Is this a common characteristic of the virion and how it is affected by presence of heparin?

Reply: The "deformation" in the force-displacement curve represents the total deformation of both the indented particle and the cantilever. Therefore, the force-displacement curve is sometimes converted to a force-indentation curve to directly read out the deformation of the indented particle during AFM nanoindentation (see also https://doi.org/10.1007/978-1-4939-8894-5_14). We now realised that the latter will be more convenient for the reader so we now display a force-indentation curve in the new figure 2B. The average particle deformation at the end of the linear phase is around 3-6 nm. As heparin engagement decreased both the viral spring constant and the critical force corresponding to the onset of non-linear deformation, the heparin addition yields no significant effect on this deformation, as shown in the following box plot.

- 7) The authors show in the infectivity experiments that dp40-highS and lower concentrations of heparin also influence the HPV16 activation. To prove the effect of the chain length and the progressive changes induces in the virus, AFM measurements of the spring constant of HPV capsids under lower concentration of heparin and dp40-highS should be provided. The same concentrations as those used in figure 1 should be tested by AFM to validate the hypothesis that the change in infectivity following an increase in heparin concentration comes from a

progressive change in the mechanical stability of viral capsids.

Reply: We appreciate the reviewer's suggestion to further investigate the effect of heparin concentration on the mechanical properties of HPV capsids. In response, we performed additional AFM measurements using the same concentrations of heparin as those used in Figure 1 to treat the HPV particles. The results demonstrate that heparin shows a similar dose-dependent effect on changing the mechanical properties of HPV. The new data have been incorporated into our revised manuscript Figure 2.

- 8) The authors state that “non-linear deformations were reversible in our nanoindentation experiments.” While I agree with these results in the linear regime, non-linear forces and the value of critical force is different between consecutive approaches, as shown in figure S2.

Reply: We appreciate the reviewer's feedback regarding the changes in critical force over indentations. The variations in critical force over successive indentation cycles can be attributed to the variance in data we generally obtain. However, the overall reversibility of the HPV particles is supported by the absence of significant changes in their morphology and size following indentation. To address these points more clearly, we have revised the relevant sentences in the manuscript to better clarify the reversibility of HPV particles over indentation cycles and to explain the observed changes in critical force. The revised text now reads: Line 345-351 “Notably, the indentation-induced non-linear deformations were reversible in our nanoindentation measurements. There were no significant changes in the morphologies and sizes of HPV16 VLPs before and after indentation (Figure S3A-B), indicating a high degree of particle robustness and flexibility withstanding loading forces up to 3 nN. The force curves corresponding to five successive indentations of a single particle showed that while the critical force slightly varied over five indentations, the slopes of each indentation were almost overlapping (Figure S3C).”

- 9) In figure 3C and 3D the data presented is the same as in figure 2D and 2E. The authors should consider rearranging the way information is presented to avoid unnecessary repetition of results within different figures.

Reply: We appreciate the reviewer's suggestion. We have incorporated the data from the original Figure 2 into the current Figure 3. Additionally, we have updated Figure 4 to ensure it does not repeat results from the current Figure 2.

- 10) During the measurement of the spring constant of the mutated HPV16, the spring constant of the mutants is presented with only 21 points. More independent experiments might be required to support the results presented.

Reply: We acknowledge the reviewer's concern regarding the number of data points for the measurement of mutant HPV16. To address this, we have conducted additional measurements, increasing the number of data points of mutant HPV to 34. Importantly, increasing the number of data points does not alter our previous conclusion that heparin has no significant effects on the size and stiffness of the mutant HPV (Figure 4B-C).

- 11) In fig 6 no labels appear in the picture. The authors should also improve the scheme's quality and add additional information to facilitate understanding of the model presented. In addition, the third panel is not necessary for the presentation of the model and should be removed from the main text.

Reply: Thank you for your suggestion. We have amended the model to improve clarity and information. Besides removing the third panel, we labelled the columns and added a legend.

- 12) In the supplementary material, the cross section along the diagonal of the virus is provided. The measurement of the cross section along the direction of the scan should be provided instead, as it is less prone to be affected by imaging artifacts. It would also be interesting to provide the height of the virus as a separate measurement showing the change in height under the presence of the HS. As mentioned above, more data and statistical analysis should be presented here to assess the relevance of these data.

Reply: We appreciate the reviewer's insightful suggestions. The diagonal line was originally taken for drawing the profile in Figure S3. We have now corrected this according to the reviewer's recommendation, providing the measurement of the cross-section along the direction of the scan to minimize the potential impact of imaging artefacts. Additionally, the height measurements of the particles under different experimental conditions have been moved from the supplementary figures to the main text to give it a more prominent place. Finally, we have added a supplementary table with mean values of relevant parameters

REVIEWERS' COMMENTS

Reviewer #1 (Remarks to the Author):

The authors have responded adequately my all of my concerns.

Reply: Thank you for your review and the positive feedback on our revision. We are pleased to hear that our responses have adequately addressed all of your concerns.

Reviewer #2 (Remarks to the Author):

The authors have addressed my main scientific concerns. My only remaining issue is with the way the summarized data have been made available. Zenodo requires a login to access the pertinent data, which for the purpose of manuscript review is not appropriate. It would essentially reveal my identity to the authors. I applaud all efforts to make more data available, but in this case (and in the future) the authors should simply upload a summary of the HDX-MS results with the manuscript or deposit the data where it is readily available without credentials. I am only specifically referring to the spreadsheet with the summarized results and sampling conditions as outlined in the Nature Methods HDX guidelines paper. File size should not be a factor, as the summarized results of even the largest HDX-MS data sets would not exceed tens of megabytes.

Reply: We are pleased to hear that our revision has addressed your main scientific concerns. We are sorry that you were not able to access the data. The direct link supplied was supposed to work without any login. To ensure that there is no problem with accessing the data now, we have made them fully open already, and also include the excel sheets as extra material for review.

Reviewer #3 (Remarks to the Author):

There are no further comments.

Reply: Thank you for your time and for acknowledging our revisions.

Reviewer #4 (Remarks to the Author):

I would like to congratulate the authors on the excellent revisions made to the manuscript. The additional experiments and thoughtful revisions have significantly enhanced the quality and clarity of the paper.

I do, however, have a few minor suggestions for further improvement.

1/ In Figure 1, I noticed that the reference column for relative infection is clearly

presented in the reply to the reviewers, but it should also be included in the article itself. Additionally, I recommend that the authors use the same normalization method for all datapoints in this figure and display the error used for normalization.

2/ In Figure 2D, the new experiments measuring the spring constant at varying heparin concentrations provide valuable insights and nicely illustrate the gradual change in the spring constant. However, I had also requested that this measurement be conducted using dp40-highS, but the results for this are missing.

Aside from these points, I find the paper to be of high quality, with interesting results that offer promising perspectives for future research.

Reply: Thank you for your positive feedback and for recognizing our revisions. We greatly appreciate your valuable insights, which have helped enhance the quality of our manuscript. For your remaining two points:

(1) As per your suggestion, we have now included the reference column (infection of untreated HaCaT cells with HPV16 PsV not incubated with heparin) in Figure 1b-e using relative infection as percentages of the reference. All datapoints in this figure were already normalized the same way, i.e. to the infection of untreated HaCaT cells with HPV16 PsV not incubated with heparin. Additionally, we have displayed the error used for normalization in the reference column.

(2) Regarding the additional measurements on dp40-highS: To our understanding, the reviewer's concerns focused on "the effect of the chain length and the progressive changes induces in the virus" and asked for "AFM measurements of the spring constant of HPV capsids under lower concentration of heparin and dp40-highS". We have (i) addressed the concern regarding the progressive changes of heparin with new AFM experiments where we tested PsVs treated with additional heparin concentrations. In addition, we have (ii) addressed the concern regarding the effect of the chain length by including the use of a short heparin oligosaccharide (Fondaparinux). Thereby we demonstrated that the chain length plays a crucial role in HPV structural activation. As shown in Figure 1, short chain lengths do not induce structural activation in HPV, thereby addressing the effects of chain length as requested. This fits with the AFM data in figure 3. With the extra measurements that we provided, we have, in our view, fully and successfully addressed the concerns of the reviewer (concerns on "the effect of the chain length and the progressive changes induces in the virus"). The addition of experiments using dp40-highS will not generate more accurate information on chain length, since – as we already pointed out in the manuscript – the source material is disperse in length and sulfation with an average dp40 and a 'high degree' of sulfation but no detailed information on the degree of dispersity. In fact, size defined heparin oligomers of dp40 are not available to anyone in the field due to the experimental constraints in purifying such material, and using this oligomer will not substantiate precisely the needed length to induce structural activation. So, we disagree with the notion that performing additional experiments on dp40-highS would be needed to address whether HS oligomers or

polymers can induce structural activation. This is the main reason why we did not include it. However, we agree that we could have explained this better in our response and we apologize for that. We hope that with the current extra explanation this point is now clarified.

Reviewer #5 (Remarks to the Author):

Reply: Thank you for your time and for co-reviewing our manuscript.